# Endosomal PI(3)P regulation by the COMMD/CCDC22/CCDC93 (CCC) complex controls membrane protein recycling

Amika Singla [1,5], Alina Fedoseienko[2,5], Sai S.P. Giridharan[3], Brittany L. Overlee[2], Adam Lopez[1], Da Jia [4], Jie Song[1], Kayci Huff-Hardy[1], Lois Weisman[3], Ezra Burstein [1,6] & Daniel D. Billadeau[2,6]

Protein recycling through the endolysosomal system relies on molecular assemblies that interact with cargo proteins, membranes, and effector molecules. Among them, the COMMD/CCDC22/CCDC93 (CCC) complex plays a critical role in recycling events. While CCC is closely associated with retriever, a cargo recognition complex, its mechanism of action remains unexplained. Herein we show that CCC and retriever are closely linked through sharing a common subunit (VPS35L), yet the integrity of CCC, but not retriever, is required to maintain normal endosomal levels of phosphatidylinositol-3-phosphate (PI(3)P). CCC complex depletion leads to elevated PI(3)P levels, enhanced recruitment and activation of WASH (an actin nucleation promoting factor), excess endosomal F-actin and trapping of internalized receptors. Mechanistically, we find that CCC regulates the phosphorylation and endosomal recruitment of the PI(3)P phosphatase MTMR2. Taken together, we show that the regulation of PI(3)P levels by the CCC complex is critical to protein recycling in the endosomal compartment.

[1] Department of Internal Medicine, and Department of Molecular Biology, University of Texas Southwestern Medical Center, Dallas, TX 75390, USA. [2] Division of Oncology Research and Department of Biochemistry and Molecular Biology, College of Medicine, Mayo Clinic, Rochester, MN 55905, USA. [3] Life Sciences Institute, University of Michigan, Ann Arbor, MI 48109, USA. [4] Key Laboratory of Birth Defects and Related Diseases of Women and Children, Department of Paediatrics, Division of Neurology, West China Second University Hospital, State Key Laboratory of Biotherapy and Collaborative Innovation Center of Biotherapy, Sichuan University, Chengdu 610041, China. [5] These authors contributed equally: Amika Singla and Alina Fedoseienko. [6] These authors jointly supervised: Ezra Burstein and Daniel D. Billadeau. Correspondence and requests for materials should be addressed to E.B. (email: ezra.burstein@utsouthwestern.edu) or to D.D.B. (email: billadeau.daniel@mayo.edu)

The levels of cell surface receptors and transporters in eukaryotic cells are critical for maintaining major physiological functions, including uptake of nutrients, electrolyte transport, growth factor signaling, cell migration, and cell adhesion. Once internalized, these plasma membrane proteins can be delivered to lysosomes for degradation, transported to the trans-Golgi network (TGN) or recycled back to the plasma membrane[1]. This tightly orchestrated process involves multiple proteins and protein complexes that are recruited to endosomal membranes through either protein–protein or protein–lipid interactions. For example, Rab5-GTP accumulation on early endosomes leads to the recruitment of multiple effector proteins including the phosphatidylinositol-3 kinase, VPS34, which converts phosphatidylinositol (PI) to PI(3)P[2–4]. Significantly, PI(3)P accumulation on endosomes facilitates the spatial and temporal recruitment of numerous proteins that contain PI(3)P-binding domains such as sorting nexins (SNX)[5]. The conversion of PI(3)P to other PI species leads to the displacement of PI(3)P-binding proteins and the recruitment of a new set of PI-binding proteins, a process that regulates endosomal maturation and receptor trafficking[1,6]. Significantly, inherited mutations in genes that regulate PI conversion in the endolysosomal system are linked to several human diseases[7–10].

Establishing different trafficking pathways for internalized receptors depends on motifs within their cytoplasmic sequences that can segregate these potential cargoes through their recognition by specific trafficking pathway regulators. Among these cargo selection regulators are members of the SNX protein family, including SNX27 and SNX17[11,12]. In addition to being recruited to endosomes through their PI(3)P-binding phox-homology domains, SNX17 and SNX27 interact with retriever and retromer, respectively, two ancient evolutionary conserved endosomally localized complexes responsible for rescuing distinct cargoes from lysosomal degradation and routing them for recycling to the cell surface or other intracellular destinations[13]. Retromer consists of VPS26A or B, VPS35 and VPS29 and is recruited to endosomes by Rab7a-GTP and SNX3[14]. VPS26A/B interacts with the PDZ domain of SNX27, which recognizes cargo proteins with PDZ-binding motifs[15], including β2 adrenergic receptor, the glucose transporter GLUT1, the copper transporter ATP7A and glutamate receptors to name a few[16–19]. Retriever is a recently identified complex composed of VPS26C (DSCR3), VPS35L (C16orf62), and VPS29[20]. Although VPS26C and VPS35L are homologous to the retromer subunits, VPS26A/B and VPS35, respectively, only VPS26C interacts with the C-terminal tail of SNX17. This factor in turn uses its FERM domain to couple to NPxY/NxxY motif-containing cargo proteins such as β1 integrin, EGFR, LDLR, and LRP1 to ensure recycling back to the plasma membrane[21–26].

Previous studies have identified that the CCC protein complex, which consists of CCDC22, CCDC93, and any of the 10 COMMD proteins, is required for retromer- and retriever-dependent protein trafficking[27–32]. For example, CCC deficiency in human and mouse cells causes defective recycling of SNX17-retriever cargoes such as β1 integrin, LDLR, and LRP1[20,31,32]. In the case of LDLR mistrafficking, this results in hypercholesterolemia due to lower uptake of LDL-cholesterol, the LDLR ligand, which is observed in animal models of CCC deficiency and patients with CCDC22 hypomorphic mutations[31]. The CCC complex is also essential for retromer SNX27-dependent recycling of the ATP7A copper transporter, and as a result, CCC depletion leads to alterations of copper homeostasis in animal models, such as Bedlington terriers with a naturally occurring mutation in COMMD1[29] and humans with CCDC22 mutations[27,33]. Other cargoes, such as Notch receptors, are also affected upon CCC complex disturbance, but sorting recognition pathways remain unclear[34].

Although the CCC complex has been defined as functionally necessary for both retromer- and retriever-dependent cargoes, a distinct biochemical or adaptor activity that may underlie this activity is currently unknown. Interestingly, CCC recruitment to endosomal membranes is dependent on the pentameric Wiskott-Aldrich syndrome protein and SCAR homologue (WASH) complex[27], a common regulator of both retromer- and retriever-dependent protein sorting[27,35]. Moreover, the CCC complex shows close co-evolution with the WASH complex[20,36], suggesting that CCC may have a primary role in regulating WASH activity. WASH serves as an actin nucleation-promoting factor for the ARP2/3 complex, which drives the generation of branched F-actin filaments on endosomal membranes[37–40]. Regulation of branched F-actin formation by WASH is needed for proper morphology of the endolysosomal system and endosomal trafficking of a broad range of cargoes[37,39,41–44], but any role for the CCC complex in this process is not currently known.

Herein, we report that that the CCC complex functions as a negative regulator of WASH complex recruitment to endosomes. We find that this is mediated through the modulation of PI(3)P levels on endosomal membranes via recruitment of the myotubularin-related protein-2 (MTMR2) lipid phosphatase.

## Results

**Defining the CCC and retriever interactomes.** A previously published study based on protein co-elution suggested the existence of a macromolecular complex of ~ 600 kDa in mass termed Commander that includes both CCC and retriever complex components, and inferred that these factors are part of a single multiprotein assembly[45]. We have previously reported that the CCC complex regulates the trafficking of cargoes that are retriever-independent, such as ATP7A, TGN46, and CIMPR[27], suggesting that CCC and retriever are functionally distinct. To try to understand if CCC and retriever are indeed separable and distinct molecular assemblies we began by defining the interactomes of CCC and retriever subunits, focusing on CCDC93 (a CCC subunit) and VPS26C (a retriever component). First, the CCDC93 or VPS26C genes were targeted in HeLa cells using CRISPR/Cas9, followed by re-expression of HA-tagged CCDC93 or VPS26C, respectively. HA-CCDC93 or HA-VPS26C immunoprecipitates were isolated and subjected to trypsin digestion for proteomic identification by mass spectrometry (data set available at MassIVE, MSV000084217 [https://massive.ucsd.edu/ProteoSAFe/dataset.jsp?accession=MSV000084217]). Factors that were isolated with stoichiometries within 10-fold from each of the baits were plotted in a Venn diagram (Fig. 1a). All components of both CCC and retriever, namely all 10 COMMD proteins, CCDC22, CCDC93, VPS35L, VPS26C, and VPS29, co-purified with each bait. These results included an additional factor, FAM45A, which was isolated by both baits (Fig. 1a). Immunoprecipitation and immunoblotting confirmed the overlapping interactions depicted in the Venn diagram (Supplementary Fig. 1a). In addition, the binding of FAM45A to CCC and retriever subunits was confirmed by co-immunoprecipitation of endogenous FAM45A with COMMD1, CCDC22, and VPS35L (Supplementary Fig. 1b). Furthermore, interaction experiments by unbiased screens in mammalian cells[46], as well as protein–protein interaction studies in Drosophila[47–49], produced interactome data that in aggregate includes all the proteins identified experimentally in our mass spectrometry data, including the identification of FAM45A as a CCC and retriever binding protein (Supplementary Data 1 and Supplementary Fig. 1c).

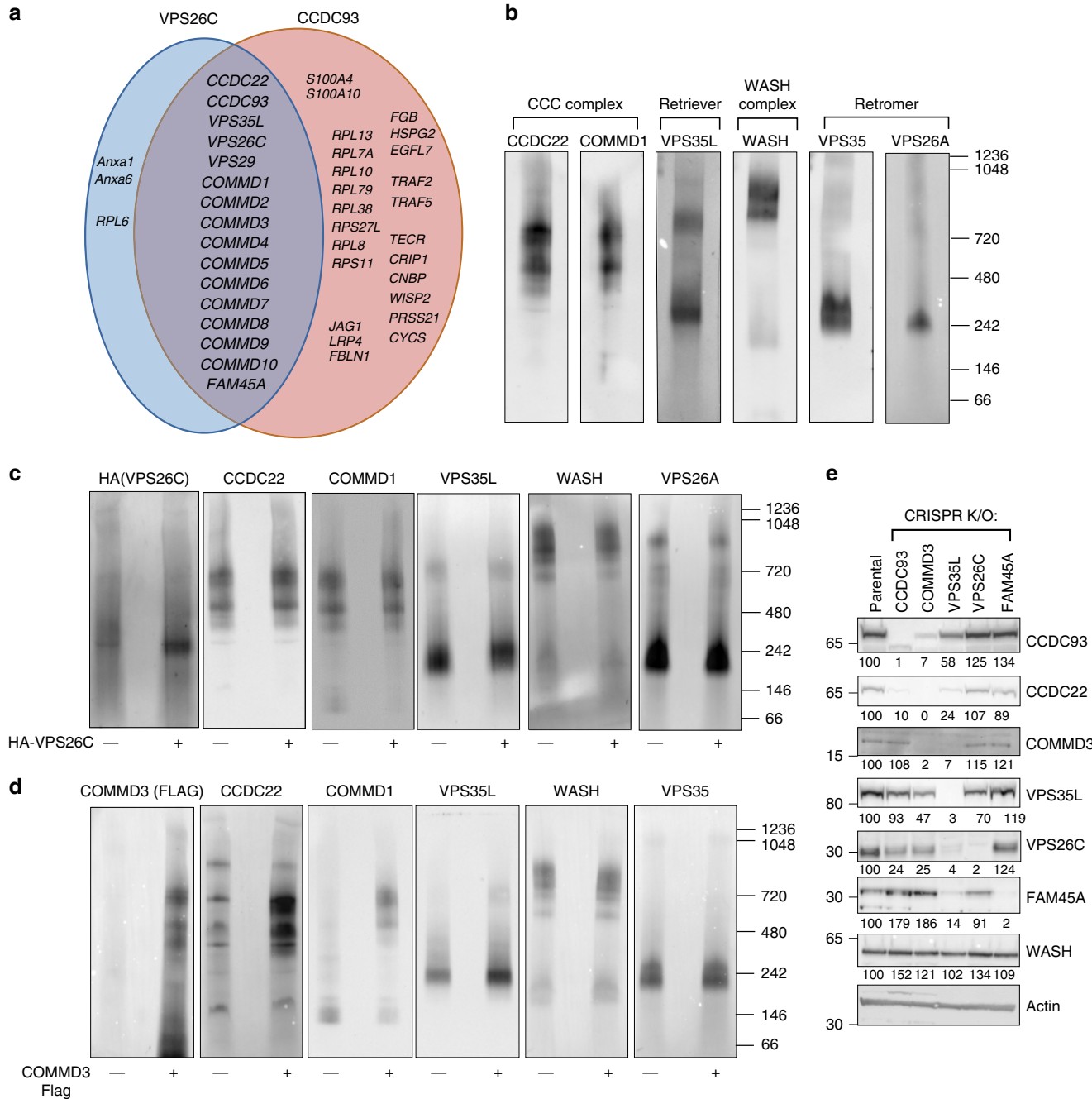

**Fig. 1** CCC and retriever are closely associated but distinct complexes. **a** Venn diagram depicting proteins identified by LC/MS-MS as interacting partners of VPS26C or CCDC93 in HeLa cells. All proteins presented were within 10-fold in abundance compared with the bait proteins. CRIPSR/Cas9 was used to make KO lines, which were then stably transduced with an empty vector (EV) or HA-tagged versions of VPS26C or CCDC93. **b**–**d** Cell lysates from parental HeLa cells **b**, VPS26C deleted or rescued cells **c**, and COMMD3 deleted or rescued cells **d** were resolved using coomassie blue native (BN) gel electrophoresis. This was followed by membrane transfer and immunoblotting using the indicated antibodies. **e** Expression of CCC and retriever complex subunits was examined by immunoblotting in the indicated HeLa CRISPR/Cas9 knockout cell lines; quantification after normalization by the loading control (Actin), is also presented as % compared with the parental control (representative of at least three separate iterations)

**CCC and retriever are closely linked but distinct complexes.**
Although these results may suggest a large single complex containing both CCC and retriever subunits, endogenous immunoprecipitation of the CCC complex (by CCDC93 or CCDC22 antibodies) led to limited or undetectable VPS26C and VPS29 coprecipitation, whereas retriever precipitation (by VPS35L antibodies) led to reliable precipitation of other retriever components (Supplementary Fig. 1d). Thus, these data suggested that CCC and retriever are closely linked but distinct assemblies. To test this

hypothesis, we used blue native (BN) gel separation and immunoblotting to ascertain the size of these molecular complexes. CCC components (COMMD1 and CCDC22) migrated close to 700 kDa, whereas VPS35L migrated in two complexes, one of similar mass ~700 kDa and one of ~250 kDa (Fig. 1b). This smaller complex was of similar mobility as retromer (VPS35 and VPS26A), and we speculated that it likely represents retriever. To examine this notion, we used VPS26C knockout and "rescue" cells (used in Fig. 1a) and observed that VPS26C is largely

restricted to this lower mass complex (Fig. 1c, VPS26C blot). Moreover, the retriever complex is of slightly lower mass in VPS26C-deficient cells compared with rescue cells, indicating that this is the main complex containing VPS26C (Fig. 1c, VPS35L blot). This behavior for the smaller VPS35L-containing complex, along with its similar mass to that of retromer, indicated to us that this likely represents retriever. In contrast to these findings, using COMMD3 knockout and "rescue" cells, we found that COMMD3 loss led to the collapse of the 700 kDa complex, best noted by the change in COMMD1 mobility and the disappearance of a 700 kDa band of VPS35L, without the loss of the 250 kDa retriever complex (Fig. 1d). Thus, we surmised that the 700 kDa complex likely represents the CCC complex and that VPS35L is a shared subunit present in both CCC and retriever. Finally, neither VPS26C or COMMD3 loss affected WASH or retromer mobility, indicating that these are separate assemblies from CCC or retriever.

Examination of these complexes was complemented by evaluating the stability of various subunits upon CRIPSR/Cas9-mediated gene deletion. Consistent with the loss of the CCC complex in COMMD3-deficient cells observed by BN gel electrophoresis (Fig. 1d), we noted that COMMD3 deletion led to significant reduction in two other CCC components, CCDC93 and CCDC22 (Fig. 1e). Moreover, just as COMMD3 deficiency had a lesser impact on retriever complex stability by BN gel electrophoresis, this corresponded to a smaller effect on VPS35L and VPS26C protein expression, as previously observed[20,27]. Similar findings were noted upon deletion of the CCC subunit CCDC93 (Fig. 1e). Conversely, VPS26C and FAM45A had minimal impact on CCC or retriever subunit expression, indicating that neither protein is required for the stability of these assemblies, and in agreement with the BN gel electrophoresis analysis of VPS26C-deficient cells (Fig. 1c). Finally, VPS35L deletion led to reduced expression of CCC subunits (CCDC22 and COMMD3 in particular) as well as retriever subunits (VPS26C). Altogether, these results are consistent with CCC and retriever being distinct entities that share VPS35L as common subunit.

**CCC links retriever to WASH-containing endosomal domains.** Although the data indicated that CCC and retriever are distinct complexes, the specific contribution of CCC to cargo trafficking remained unclear. Previous studies relying on siRNA[20] suggested that the CCC complex is important for the recruitment of retriever to endosomes. We tested that notion by comparing VPS35L endosomal recruitment in parental HeLa cells and COMMD3-deficient HeLa cells that as noted above lose CCC complex assembly but retain retriever (albeit in somewhat reduced levels). Interestingly, deletion of COMMD3 led to loss of VPS35L endosomal localization, in contrast to parental wild type (WT) HeLa, which displayed VPS35L puncta near EEA1- and retromer-containing endosomal domains in (Fig. 2a). Next, we sought to examine the mechanism by which the CCC complex might mediate retriever recruitment to endosomes. Previously, it was shown that the endosomal recruitment of the CCC complex is mediated by the WASH complex subunit FAM21 through direct interactions with CCDC93[27]. Given the loss of endosomal localization of VPS35L in COMMD3-deficient cells, we hypothesized that retriever localization would be dependent on WASH complex interactions mediated by the CCC complex. Using COMMD3-deficient and "rescue" cells, we immunoprecipitated VPS35L and examined its interaction with the WASH complex. Consistent with the hypothesis, VPS35L binding to WASH was dependent on COMMD3 expression and was absent in COMMD3-deficient cells (Fig. 2b). Taken together, these data

indicate that the CCC complex provides a link between the retriever and WASH complexes, and this link is required for endosomal recruitment of retriever.

**CCC loss leads to retriever-independent recycling defects.** If the sole function of the CCC complex is to link retriever to endosomes, it would be predicted that its deficiency should phenocopy retriever loss. However, according to prior studies[27,34], a number of retriever-independent cargos were found to also be mislocalized upon silencing of selective CCC complex subunits. Using HeLa cells generated through CRISPR/Cas9 genome editing, we examined the trafficking of a retromer cargo protein, GLUT1, in CCC-deficient (COMMD3 and CCDC93), retriever-deficient (VPS35L, VPS26C), and FAM45A-deficient cells. Deletion of CCC, VPS35L and FAM45A, resulted in a pronounced accumulation of GLUT1 in early endosomes, consistent with the notion that the CCC complex has broader functions than retriever regulation (Fig. 2c, d). Only VPS26C deletion had no effect on GLUT1 trafficking, in line with the preservation of CCC in these cells and the previous report that VPS26C does not regulate trafficking of SNX27-dependent cargoes, like GLUT1[20].

Next, we analyzed in detail the mechanisms of mistrafficking of integrin β1 (ITGβ1), a retriever cargo, also sensitive to CCC complex depletion[20–22]. McNally and coauthors had reported that upon transient knockdown of CCDC22, CCDC93 and VPS35L, integrin α5/β1 surface levels are decreased and the protein is routed for lysosomal degradation[20]. First, we examined ITGβ1 recycling in a panel of HeLa knockout cell lines deficient in CCC, retriever or FAM45A. In this assay, where the plasma membrane pool of ITGβ1 is labeled and its localization monitored over time, ITGβ1 was predominantly at the plasma membrane in WT HeLa cells (1 h after initial labeling). In contrast, deletion of CCC (COMMD3, CCDC93), VPS35L and VPS26C led to pronounced accumulation of ITGβ1 on WASH-positive intracellular vesicles (Supplementary Fig. 2a, b). Interestingly, FAM45A did not affect trafficking of this receptor. In line with further mistrafficking to lysosomes, ITGβ1 protein levels were reduced in every instance when endosomal trapping was observed, and these reduced levels were restored upon treatment with the lysosome protease inhibitor Bafilomycin A (Supplementary Fig. 2c). The accumulation of ITGβ1 on WASH-positive vesicles in CCC knockout cells was not due to increased endocytosis, which appeared unaffected (Supplementary Fig. 2d), but due to defective endosomal recycling (Supplementary Fig. 2e, f). Taken together, these data indicate that the CCC complex can regulate endosomal recycling of both retromer- and retriever-dependent cargoes.

**CCC depletion results in F-actin accumulation on endosomes.** Components of the CCC complex are highly conserved and demonstrate strong co-evolution with the WASH complex[20,36], suggesting a regulatory role in WASH activity. This possibility, along with the observed broad repercussions of CCC loss on protein recycling, led us to investigate whether CCC might regulate WASH complex activity. First, we assessed endosomal F-actin in CCC- and retriever-deficient cells using fluorescently labeled phalloidin (Fig. 3a). The staining revealed accumulation of F-actin on WASH-positive endosomal foci in CCDC93, COMMD3, and VPS35L KO cells, but not in VPS26C and FAM45A KO cells (Fig. 3b). Significantly, *Commd1*-deficient mouse embryonic fibroblasts and fibroblasts derived from a patient with a *CCDC22* hypomorphic mutation (*CCDC22* p. T17A) also showed increased F-actin accumulation on endosomes (Supplementary Fig. 3a, b). We also observed accumulation of cortactin, an ARP2/3 complex and F-actin-stabilizing

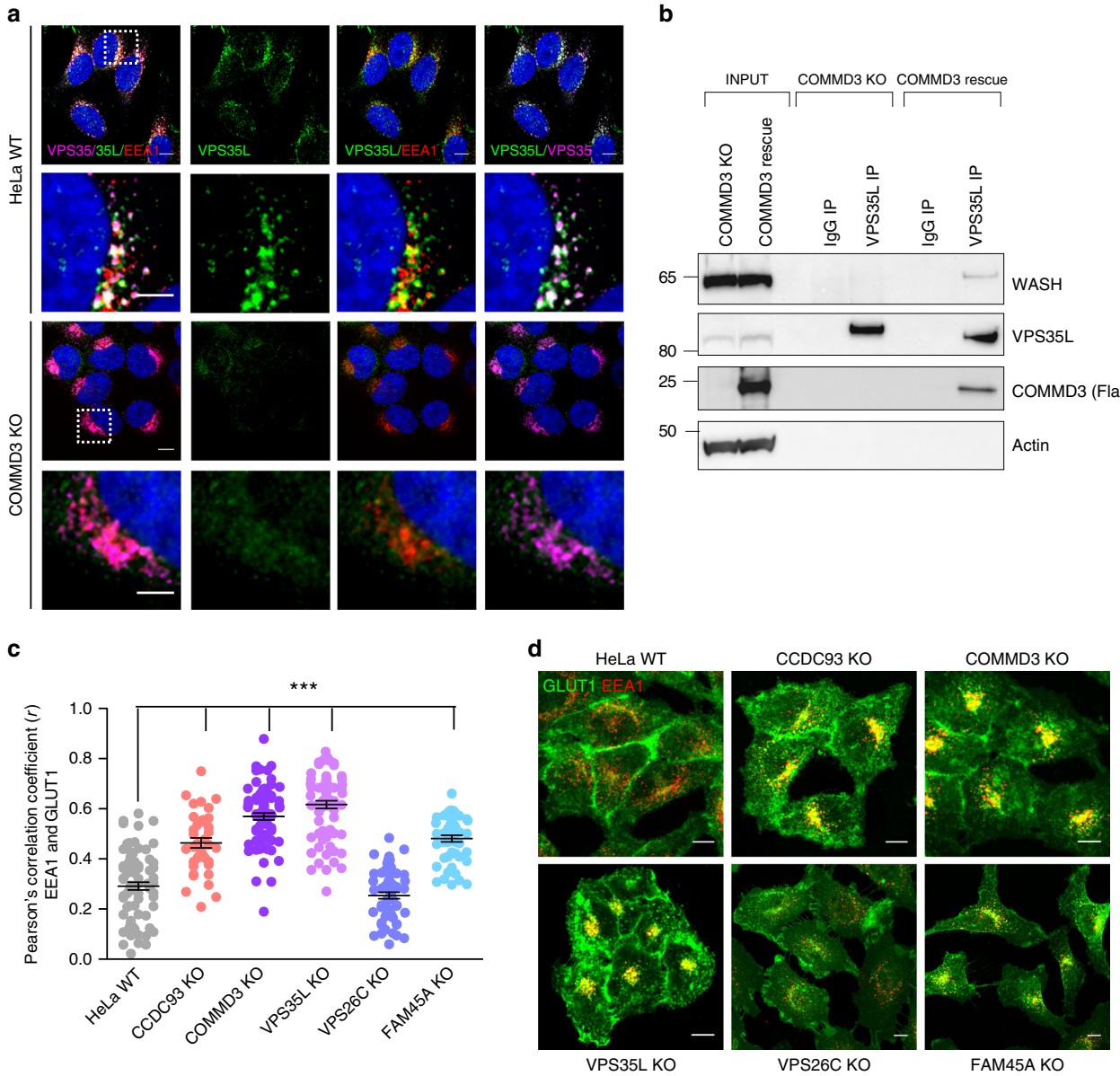

**Fig. 2** CCC links retriever to endosomes but its loss affects retriever-independent cargoes. **a** Confocal images from VPS35L, EEA1, and VPS35 immunofluorescence staining depicting the subcellular localization of the retriever subunit VPS35L in the indicated cell lines. Representative images of two independent experiments are shown. Scale bars, 10 μm, for zoomed images 5 μm. **b** Immunoprecipitation of the retriever subunit VPS35L was followed by immunoblotting for WASH. Lysates from COMMD3 KO cells and an isogenic rescue line were used (representative of two independent iterations). **c** Pearson correlation coefficients for GLUT1 and EEA1. Results for individual cells are plotted, along with the mean and s.e.m. for each group ($n = 76$ cells for WT, 76 for COMMD3 KO, 38 for CCDC93 KO, 72 for VPS35L KO, 59 for VPS26C KO, and 50 for FAM45A KO); ***$P < 0.0001$ (one-way ANOVA and Dunnett's test to control). **d** Confocal microscopy imaging of EEA1 and GLUT1 immunofluorescence staining quantified in **c**. Representative images from three independent experiments are shown. Scale bars, 10 μm

protein, on FAM21-positive vesicles in CCC-deficient HeLa cells (Fig. 3c, d). Closer observation of actin endosomal structures in CCC-deficient cells revealed the presence of WASH, FAM21, and VPS35 at the tips of cortactin tails, and ITGα5 with FAM21 at the tips of F-actin tails (Supplementary Fig. 3c). Similar F-actin-containing structures have been observed in cells when WASH activity was increased[50].

We next asked whether F-actin accumulation on endosomes in CCC-deficient cells was WASH-dependent. To that end, we assessed endosomal F-actin in CCDC93 KO cells and found that depletion of WASH (shWASH) led to near complete loss of F-actin on endosomes of these cells, whereas the CCDC93-deficient cell line infected with a control shRNA lentivirus continued to

show enhanced F-actin accumulation on endosomes, (Fig. 3e, f). Taken together, these data indicate that CCC, but not retriever, negatively regulates WASH-dependent endosomal F-actin accumulation. Once again, VPS35L behaved functionally as other components of the CCC complex.

**CCC depletion results in increased WASH complex activation.** To ascertain the mechanism of endosomal F-actin in CCC-deficient cells, we looked at WASH distribution in both CCC- and retriever-deficient cells. Immunofluorescence revealed that endosomal-associated WASH levels were increased in CCDC93, COMMD3, and VPS35L KO, but not in VPS26C KO and

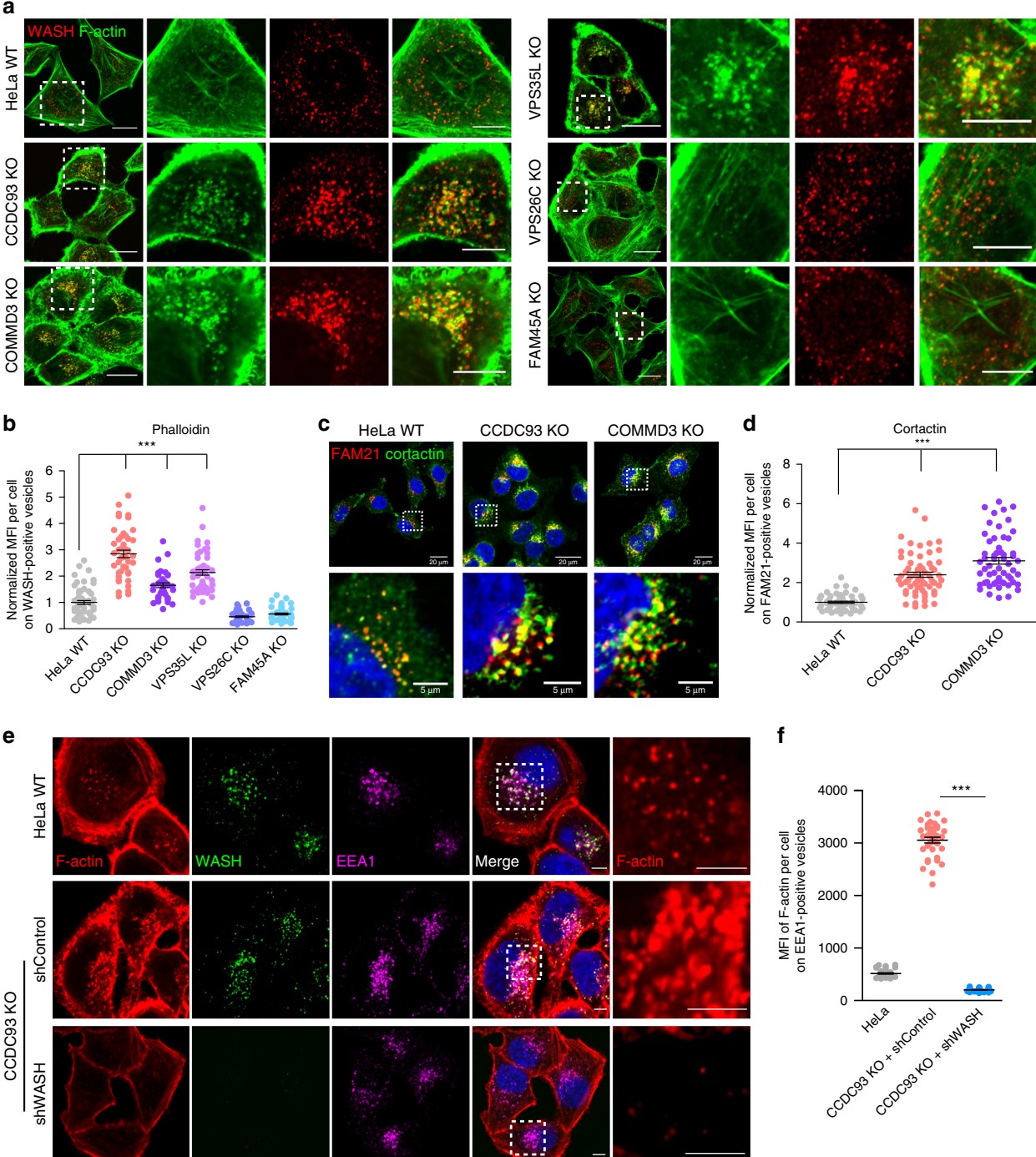

**Fig. 3** CCC depletion leads to WASH-dependent increase in endosomal F-actin. **a** Confocal imaging for F-actin and WASH immunofluorescence staining of HeLa WT and the indicated CRISPR/Cas9 knockout cell lines. Representative images of three independent experiments are shown. Scale bars 20 μm, for zoomed images 10 μm. **b** Quantification of normalized mean fluorescent intensity (MFI) of F-actin on WASH-positive vesicles from **a**. Results for individual cells are plotted, along with the mean and s.e.m. for each group ($n = 62$ cells for WT, 45 for CCDC93 KO, 48 for COMMD3 KO, 50 for VPS35L KO, 43 for VPS26C KO, 48 for FAM45A KO); ***$P < 0.0001$ (one-way ANOVA and Dunnett's test to control). **c** Confocal imaging for cortactin and FAM21 immunofluorescence staining in the indicated cell lines. Representative images of two independent experiments are shown. Scale bars 20 μm, for zoomed images 5 μm. **d** Quantification of cortactin immunofluorescence on FAM21-positive vesicles from **c**. Results for individual cells are plotted, along with the mean and s.e.m. for each group ($n = 51$ for WT cells, 63 for CCDC93 KO, 56 for COMMD3 KO); ***$P < 0.0001$ (one-way ANOVA and Dunnett's test to control). **e** Confocal imaging of F-actin, EEA1, and WASH immunofluorescence staining for the indicated cell lines. Representative images of two independent experiments are shown. Scale bars, 10 μm and 5 μm on zoomed images. **f** Quantitation of endosomal F-actin deposition on EEA1-positive endosomes in the indicated cell lines ($n = 35$ cells for HeLa WT, 36 for CCDC93 KO shcontrol, 37 for CCDC93 shWASH). The mean MFI and the s.e.m. for each group are plotted; ***$P < 0.0001$ (unpaired two-tailed $t$ test to CCDC93 KO shcontrol)

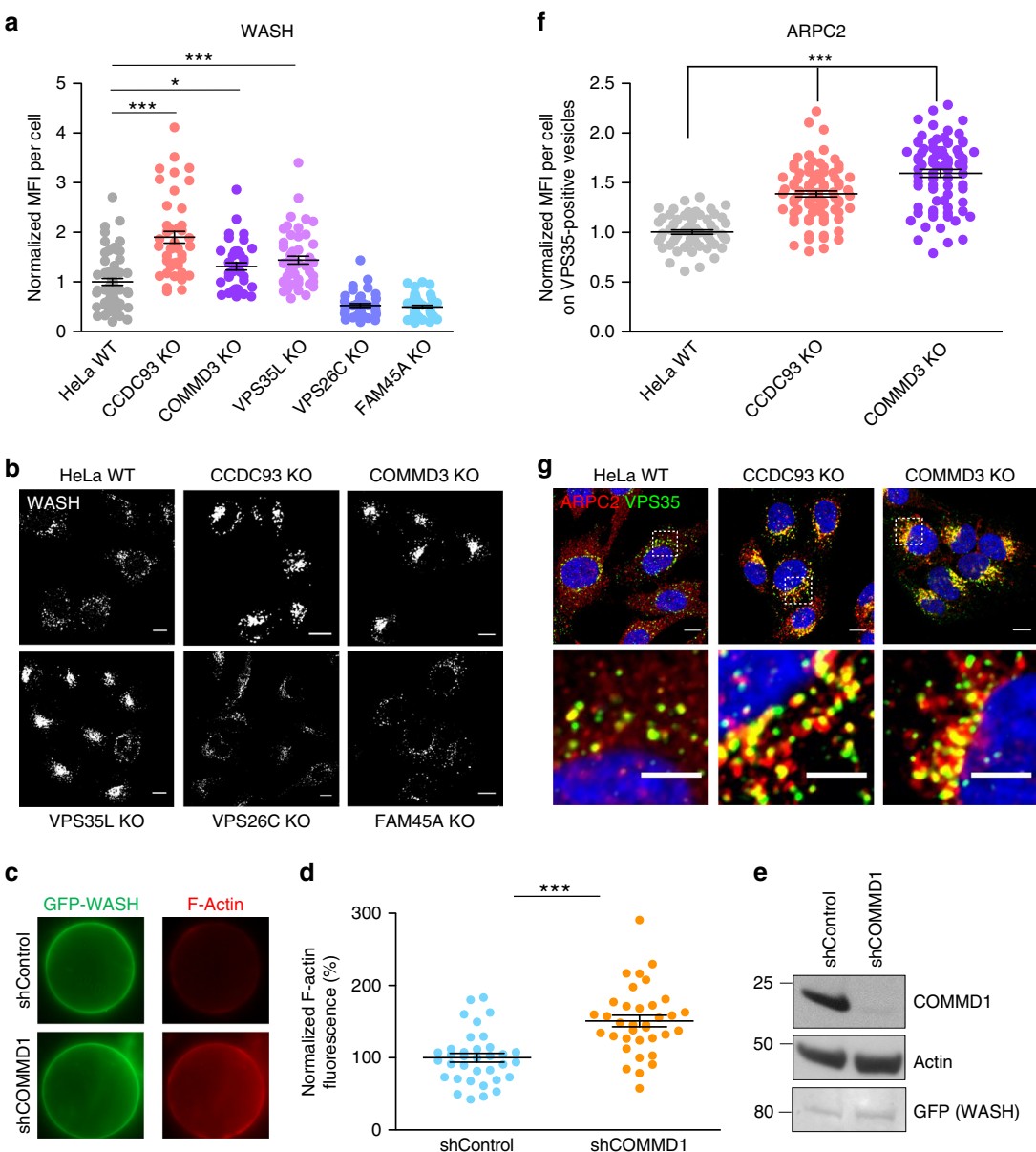

**Fig. 4** WASH endosomal recruitment and activity are increased in CCC-deficient cells. **a** Normalized mean fluorescence intensity (MFI) of WASH was quantified in the indicated cell lines. Results for individual cells are plotted, along with the mean and s.e.m. for each group ($n = 62$ cells for HeLa WT, 45 for CCDC93 KO, 48 for COMMD3 KO, 50 for VPS35L KO, 43 for VPS26C KO, 48 for FAM45A KO); ***$P < 0.0001$; *$P < 0.05$ (one-way ANOVA and Dunnett's test to control). **b** Confocal imaging of WASH immunofluorescence staining in the indicated cell lines, corresponding to **a**. Representative images are shown from two independent experiments. Scale bars, 10 μm. **c** WASH complex activity was determined by pyrene-actin assembly assays in the indicated cell lines (*Wash* knockout fibroblasts reconstituted with HA-GFP-WASH, that were further transduced with shControl or shCommd1 lentiviruses). WASH recovery and F-actin deposition on protein G beads was determined by fluorescence microscopy (examples shown on the left panels). **d** Quantification of F-actin fluorescence, normalized to WASH recovery is shown (middle panel), with mean values and s.e.m. plotted ($n = 35$ cells); ***$P < 0.0001$ (unpaired two-tailed $t$ test to control). **e** Immunoblot analysis of the two cell lines is also shown (right panel), including Commd1, WASH (GFP), and actin (as a loading control). Results are representative of three independent iterations. **f** Quantification of ARPC2 immunofluorescence on VPS35-positive endosomes. Results for individual cells are plotted, along with the mean and s.e.m. for each group ($n = 60$ cells for WT, 84 for CCDC93 KO, 77 for COMMD3 KO); ***$P < 0.0001$ (one-way ANOVA and Dunnett's test to control). **g** Confocal imaging of ARPC2 and VPS35 immunofluorescence staining in the indicated cell lines, corresponding to **f**. Representative images of two independent experiments are shown. Scale bars, 10 μm on non-zoomed and 5 μm on zoomed images

FAM45A KO cells (Fig. 4a, b). Interestingly, compared with control cells, WASH-positive vesicles in CCDC93−, COMMD3−, and VPS35L-deficient cells appeared substantially enlarged and accumulated in a tighter perinuclear area (Fig. 4b), thus resulting in a smaller area of the cytosol being occupied by WASH-positive vesicles (Supplementary Fig. 4a).

Next, we assessed WASH-mediated actin-nucleating activity. WASH complexes were immunoprecipitated from *Wash* knock-out mouse embryonic fibroblasts reconstituted with HA-tagged GFP-WASH. CCC depletion through *Commd1* silencing by shRNA led to significant increase in actin-nucleating activity (Fig. 4c–e). The same was observed regarding CCC depletion

when endogenous WASH complexes were precipitated from HEK293 cells (Supplementary Fig. 4b, c). Interestingly, WASH complex assembly seemed unaffected as co-precipitation of FAM21 with other components of this complex was indistinguishable between control and shCommd1 cells (Supplementary Fig. 4d), in agreement with the unaffected mobility of the WASH complex in CCC- and retriever-deficient cells (Fig. 1c, d).

Consistent with the increased nucleating activity, we found that CCC-deficient cells displayed increased Arp2/3 recruitment to retromer-positive vesicles, as demonstrated by staining for the Arp2/3 subunit ARPC2 (Fig. 4f, g). Moreover, ARPC2 foci were substantially larger in COMMD3-deficient cells, where they still contained internalized ITGα5, which in control conditions was not trapped on endosomes (Supplementary Fig. 4e). Altogether, our data indicate that the CCC complex functions as a negative regulator of WASH-dependent F-actin deposition, and suggests that this negative regulation is required for optimal cargo sorting from endosomes.

**CCC deficiency results in endosomal accumulation of PI(3)P.**
Phosphoinositide (PI) levels are tightly regulated and play an essential role in vesicular budding, membrane fusion, and cytoskeleton organization[51,52]. In fact, increases in several PI species have been associated with the accumulation of endosomal F-actin[9,50,53,54], which prompted us to investigate whether PI metabolism was impaired in CCC-deficient cells. As PI(3)P plays a central role in endosomal recruitment of effector proteins, such as early endosomal antigen 1 (EEA1), we assessed the levels of endosomally localized EEA1 in CCC- and retriever-deficient cells. Interestingly, endosomal EEA1 levels were significantly elevated in CCDC93 and COMMD3 KO cells, and only slightly elevated in VPS26C KO cells (Fig. 5a, b and Supplementary Fig. 5a), a result that correlated with WASH levels on EEA1-positive endosomes in these cells (Fig. 5c). To determine if PI(3)P levels were elevated on endosomes in CCC-deficient cells, we transfected cells with a vector expressing a PI(3)P reporter protein consisting of a fusion between dsRed and the FYVE-domain from EEA1. The PI(3)P reporter signal revealed a ~2.5-fold increase in PI(3)P levels in endosomes from COMMD3 and CCDC93 KO cells, which was not seen in VPS26C KO cells (Fig. 5d, e). In order to more quantitatively measure levels of PI(3)P and other PIP species, we performed HPLC in control and COMMD3-deficient cells. As shown in Fig. 5f, COMMD3 deficiency resulted in a ~3.2-fold increase in total PI(3)P levels; other PIs were also elevated but to a lesser extent (Supplementary Fig. 5b). Taken together, these data indicate that CCC deficiency, but not retriever deficiency, alters specific PI lipids and has the largest impact on endosomal PI(3)P levels.

**VPS34 blockade reverts endosomal phenotypes of CCC depletion.** The above data indicate that CCC complex deficiency results in abnormally high levels of PI(3)P on endosomes. To test if this alteration is linked to WASH recruitment and receptor trafficking, we tested whether pharmacologic inhibition of VPS34, the main lipid kinase involved in the generation of endosomal PI(3)P[2–4], could rescue the effects induced by CCC complex loss. Indeed, chemical inhibition of VPS34 using a highly specific VPS34 inhibitor (VPS34-IN1)[55] reduced the PI(3)P reporter signal in CCC-deficient cells (Fig. 6a, b). Moreover, VPS34 inhibition significantly reduced endosomal EEA1 and WASH complex recruitment, and prevented excessive endosomal F-actin accumulation in CCDC93 and COMMD3 KO cells (Fig. 6c and Supplementary Fig. 6a–c). To ensure that these results were not due to unintended off-target effects of the VPS34 inhibitor, we used three independent VPS34 siRNAs (Supplementary Fig. 6d).

Once more, siRNA depletion of VPS34 in COMMD3-deficient cells diminished endosomal WASH recruitment and most significantly, this largely eliminated the endosomal trapping of internalized ITGα5 (Fig. 6d–f). In addition, VPS34 silencing in COMMD3-deficient led to partial restoration of endosomal morphology and increased the cellular area of WASH-positive vesicles (Supplementary Fig. 6e). These results indicate that in CCC-deficient cells, elevated WASH recruitment to endosomal structures, and its downstream consequences including increased F-actin generation and mistrafficking of receptors, are dependent on PI(3)P accumulation.

**The PI(3)P phosphatase MTMR2 is regulated by the CCC complex.** The increase in endosomal PI(3)P levels in CCC-deficient cells could be caused by either increased VPS34 activity, decreased conversion of PI(3)P into PI(3,5)P$_2$ (defective PIKfyve kinase activity), or by lower rates of dephosphorylation of PI(3)P back into PI by myotubularins phosphatases. Using an ELISA-based in vitro kinase reaction, we found that VPS34 immuno-precipitated from CCC-deficient cells was not significantly more active compared to control cells (Supplementary Fig. 7a, b). Moreover, CCC deficiency did not impact the interaction of Beclin1 with VPS34 and ATG14, which are additional components of the VPS34 kinase complex (Supplementary Fig. 7c). These data suggest that VPS34 activity is unchanged in CCC-deficient cells. Further, PIKfyve activity seemed intact as conversion of PI3P into PI(3,5)P$_2$ in CCC-deficient cells was not impaired judging by the noted increase in PI(3,5)P$_2$ (Supplementary Fig. 5b).

As VPS34 activity did not appear to be increased in CCC-deficient cells, we focused our attention on PI(3)P phosphatases, specifically myotubularins[56], which also regulate PI(3)P levels. Recently, two independent high-throughput proteomics screens identified that MTMR2 may interact with CCDC22[46,57]. We examined the possibility of a CCDC22 interaction with MTMR2 by co-immunoprecipitation experiments. First, we performed immunoprecipitation of endogenous CCDC22 and found that it readily co-immunoprecipitated HA-tagged MTMR2 (Fig. 7a). Furthermore, immunoprecipitation of endogenous CCDC22 also led to the recovery of endogenous MTMR2 (Fig. 7b). Closer examination of the interaction, indicated that a central region in CCDC22 (amino acids 321–447), preceding the C-terminal coil-coiled domain in this protein, is both necessary and sufficient for its interaction with MTMR2 (Fig. 7c). Interestingly, this region does not bind to COMMD1 and thus does not support CCC complex assembly, indicating that the binding between CCDC22 and MTMR2 does not require a fully assembled CCC complex. Conversely, we found that full-length MTMR2 is required for its interaction with CCDC22 and no single domain was able to recapitulate binding (Supplementary Fig. 8a).

Next, we examined whether MTMR2 knockdown could mimic the endosomal phenotypes observed in CCC-deficient cells. MTMR2 knockdown led to pronounced accumulation of GLUT1 on EEA1-positive endosomes compared with control siRNA-transfected cells (Supplementary Fig. 8b–d). Further, a significant accumulation of FAM21 and F-actin on endosomes was seen in MTMR2-deficient cells, akin to the phenotype of CCC deficiency (Fig. 7d, Supplementary Fig. 8e). Most importantly, MTMR2 silencing also led to defective endosomal recycling of ITGβ1, and associated WASH accumulation on endosomes, both of which were reversed by inhibition of PI(3)P synthesis (Fig. 7e, Supplementary Fig. 8f). Thus, our data indicated that MTMR2 not only interacts with the CCC complex but its deficiency phenocopies the endosomal-recycling defect seen in CCC-deficient cells.

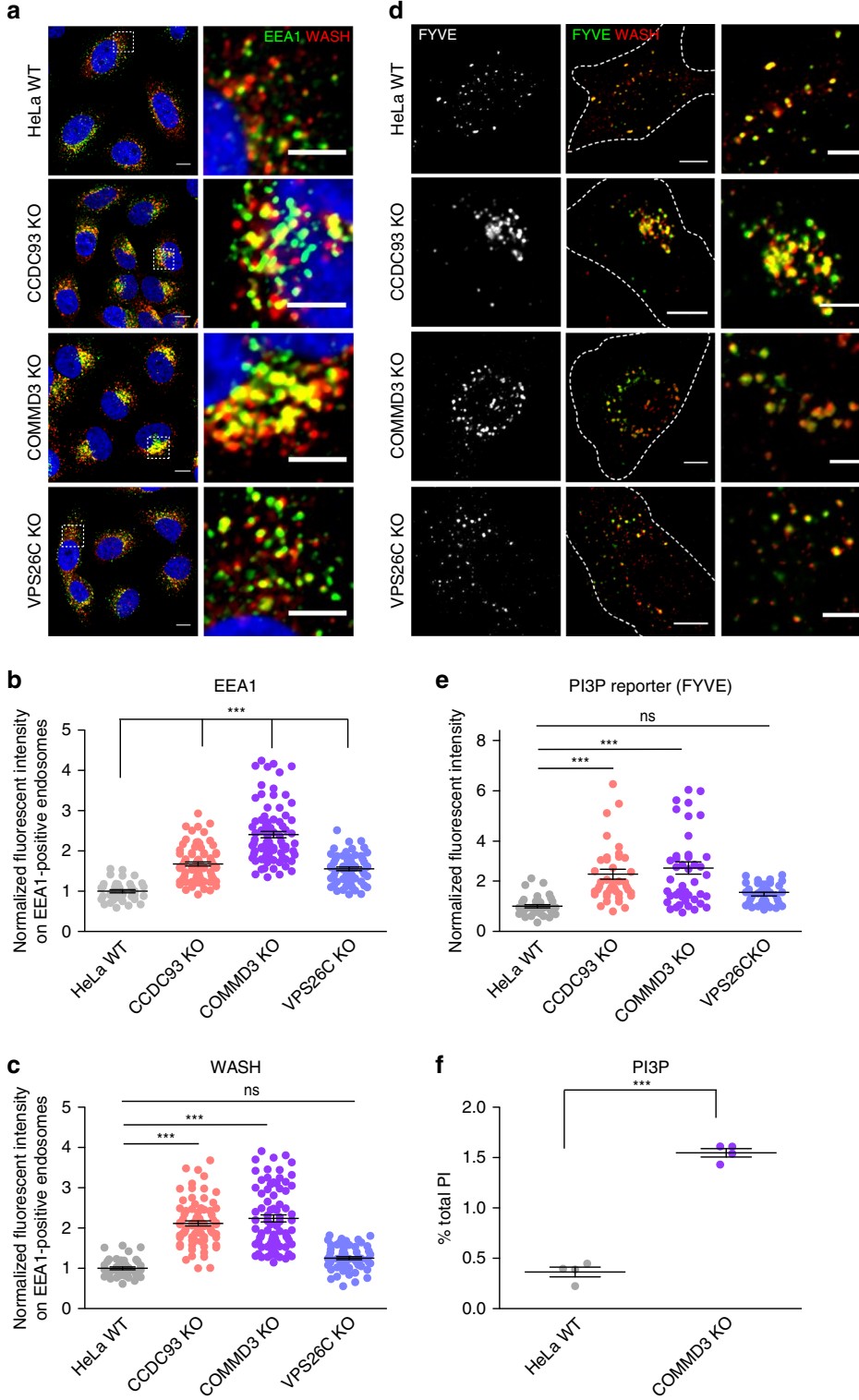

Given the physical interaction of MTMR2 with CCDC22, we investigated whether the endosomal recruitment of MTMR2 was affected by CCC deficiency. Phosphorylation of MTMR2 on serine 58 has a profound effect on its cellular localization, preventing its recruitment to endosomes[58]. This residue is preceded by phenylalanine and upon phosphorylation may be recognized by a phosphoserine antibody raised against this motif (phosphoserine preceded by tyrosine, phenylalanine, or tryptophan). Thus, to investigate the phosphorylation status of

MTMR2, we immunoprecipitated the protein and immunoblotted the resulting material with this specific phosphoserine antibody. We observed that COMMD1-deficient HEK293T cells (which display exaggerated WASH complex activity, Supplementary Fig. 4b, c), had marked elevation in phosphorylated MTMR2 (Fig. 7f), a form of the protein that has been previously shown to be unable to bind to endosomes[58]. To investigate if defective MTMR2 recruitment to endosomes is responsible for the phenotype resulting from COMMD3 deficiency, we expressed

**Fig. 5** PI(3)P levels are increased in CCC-deficient cells. **a** Confocal microscopy imaging from EEA1 and WASH immunofluorescence staining in the indicated cell lines, corresponding to **b** and **c**. Representative images of two independent experiments are shown. Scale bars, 10 µm on non-zoomed and 5 µm on zoomed images. For complete staining see Supplementary Fig. 4. **b** Quantification of the normalized mean fluorescence intensity of EEA1 in the indicated cells lines. Results for individual cells are plotted, along with the mean and s.e.m. for each group ($n = 42$ cells for WT, 69 for CCDC93 KO, 86 for COMMD3 KO, 62 for VPS26C KO); ***$P < 0.0001$ (one-way ANOVA and Dunnett's test to WT control). **c** In the same cells used for **b**, WASH mean fluorescence intensity in the region occupied by EEA1-positive endosomes was quantified; ***$P < 0.0001$; ns, not significant (one-way ANOVA and Dunnett's test to control). **d** Confocal microscopy imaging from cells transfected with a dsRed-EEA1-FYVE domain PI(3)P reporter plasmid (pseudocolored in green) and co-stained for WASH. Scale bars, 10 µm on non-zoomed and 5 µm on zoomed images. **e** Quantification of the normalized mean fluorescence intensity of the PI(3)P reporter (FYVE). Results for individual cells are plotted, along with the mean and s.e.m. for each group ($n = 35$ cells for WT, 43 for COMMD3 KO, 40 for CCDC93 KO, 36 for VPS26C KO); ***$P < 0.0001$; ns, not significant (one-way ANOVA and Dunnett's test to control). **f** PI(3)P levels determined by HPLC in HeLa WT and COMMD3 KO cells. Mean and s.e.m. of four biological replicates per group are plotted. Representative data of two independent experiments are shown; ***$P < 0.0001$ (unpaired two-tailed $t$ test to control)

HA-tagged MTMR2 wild type (WT) or phosphorylation-resistant mutant (S58A). As seen by others, MTMR2 S58A localized almost exclusively to endosomes, whereas the WT protein was largely cytosolic and its presence in endosomes was hard to detect by imaging in WT cells (Fig. 7g, top row), probably indicative that small amounts of this enzyme are ever recruited to these structures under physiologic conditions. Importantly, endosomally localized MTMR2 S58A was able to rescue the accumulation of endosomal F-actin noted in COMMD3-deficient cells (Fig. 7g, bottom row, Fig. 7h). In aggregate, the data indicate that the CCC complex regulates MTMR2 phosphorylation, a key step in its recruitment to endosomes, and that MTMR2 perturbations results in endosomal phenotypes that correspond to the functional outcomes observed upon deletion of CCC complex subunits.

## Discussion

The CCC complex is a molecular assembly that has been recognized for its role in regulating endosomal recycling of various cargos[20,27,30–32,34], which as shown here, involve both retromer and retriever pathways. However, the mechanism by which the CCC complex modulates endosomal protein trafficking remained unclear. The present study provides a detailed mechanistic analysis of this question and uncovered that the CCC complex modulates WASH recruitment and activity by regulating PI(3)P levels on endosomal membranes through the PI(3)P phosphatase MTMR2 (Fig. 7i).

Consistent with prior work, we found that CCC complex subunits interact with components of retriever, a complex that is structurally analogous to retromer. Both retromer and retriever are trimeric assemblies that share one common subunit (VPS29), and can utilize specific SNX proteins to identify cargo in the endosomal compartment[14,15,20]. Although some have suggested that CCC and retriever represent one large multiprotein complex termed Commander[59], data presented here indicate that the CCC complex and retriever are two separable assemblies of different sizes (~700 kDa and 250 kDa, respectively), which share VPS35L as a common subunit. Interestingly, in cell lysates both retriever and retromer have similar mobility, which in both cases is larger than the predicted trimeric assembly of their principal subunits, predicted at ~150–170 kDa.

In addition, we find that CCC and retriever have different functional roles as exemplified by the endosomal phenotypes observed upon selective gene deletion. In this regard, altered endosomal morphology, F-actin, and WASH accumulation, as well as PI(3)P accumulation, are unique to CCC complex deletion and not seen in VPS26C- or FAM45A-deleted cell lines. Interestingly, VPS35L deletion shares properties with CCC-deficient states consistent with its presence in both retriever and the CCC complexes. Moreover, our data indicate that the CCC complex, through the direct interaction between CCDC93 and FAM21,

provides a link between retriever and the WASH complex, which is required for endosomal recruitment (Fig. 7i). Another finding in this study is that FAM45A, a DENN-domain containing protein of unknown function, binds to CCC and retriever, in agreement with high-throughput protein–protein interaction studies[46,60]. The role of FAM45A in endosomal sorting is still unknown: while it lacks the ability of the CCC complex to regulate endosomal F-actin accumulation (suggesting that it probably does not regulate PI(3)P levels), it seems to have a more-significant effect on retromer-mediated recycling.

Evolutionary conservation analysis indicates that CCC subunits co-evolved with the WASH complex[20,36], and our data suggest that this occurred in order to regulate the endosomal recruitment and activity of this actin-nucleating factor. WASH is recruited to endosomes where it is necessary for branched F-actin deposition, a process that is essential to maintain the structure of the endolysosomal network, to promote effective fission of membrane tubules and trafficking of cargo proteins[37,39,41,43,44]. Importantly, we find that the CCC complex functions as a negative regulator of WASH, highlighting that excess endosomal F-actin accumulation is detrimental to endosomal cargo recycling. The increased endosomal recruitment of the WASH complex in CCC-deficient cells is directly correlated to an increased activity state for this complex, and to F-actin accumulation. This result is in agreement with the previously published proteomics data showing that WASH levels are increased in hepatocytes from *Commd1* and *Commd6* tissue-specific knockout mice that show defective LDLR recycling[32]. Furthermore, they are in agreement with a recent study that identified a number of alterations in cytoskeletal structure in COMMD5-deficient cells[61]. Altogether these data suggest that the enhanced endosomal accumulation of WASH in CCC-deficient states is physiologically detrimental.

Our studies indicate that WASH complex hyper-recruitment and activation, seen in CCC deficiency, occur as a result of endosomal PI(3)P accumulation. This finding adds another dimension to the importance of PI(3)P regulation on endosomes, as our data uncovered that PI(3)P turnover is necessary for the loss of WASH complex from endosomes and the termination of F-actin nucleation, without which the sorting and recycling processes do not progress normally. The precise mechanisms by which endosomal PI(3)P is involved in the recruitment and/or activation of the WASH complex remain to be fully elucidated. However, it should be noted that FAM21 has two basic regions within its extended C-terminal tail that have been shown to interact with PI(3)P and other PIs[40], and several WASP family members including WASP/N-WASP, WAVE proteins, and WHAMM, all contain PIP-interacting motifs that impact their localization and activation state[62–66].

As far as the mechanism for PI(3)P accumulation in CCC-deficient states, the aggregate of the data included here indicate that the main defect is in PI(3)P conversion to PI owing to

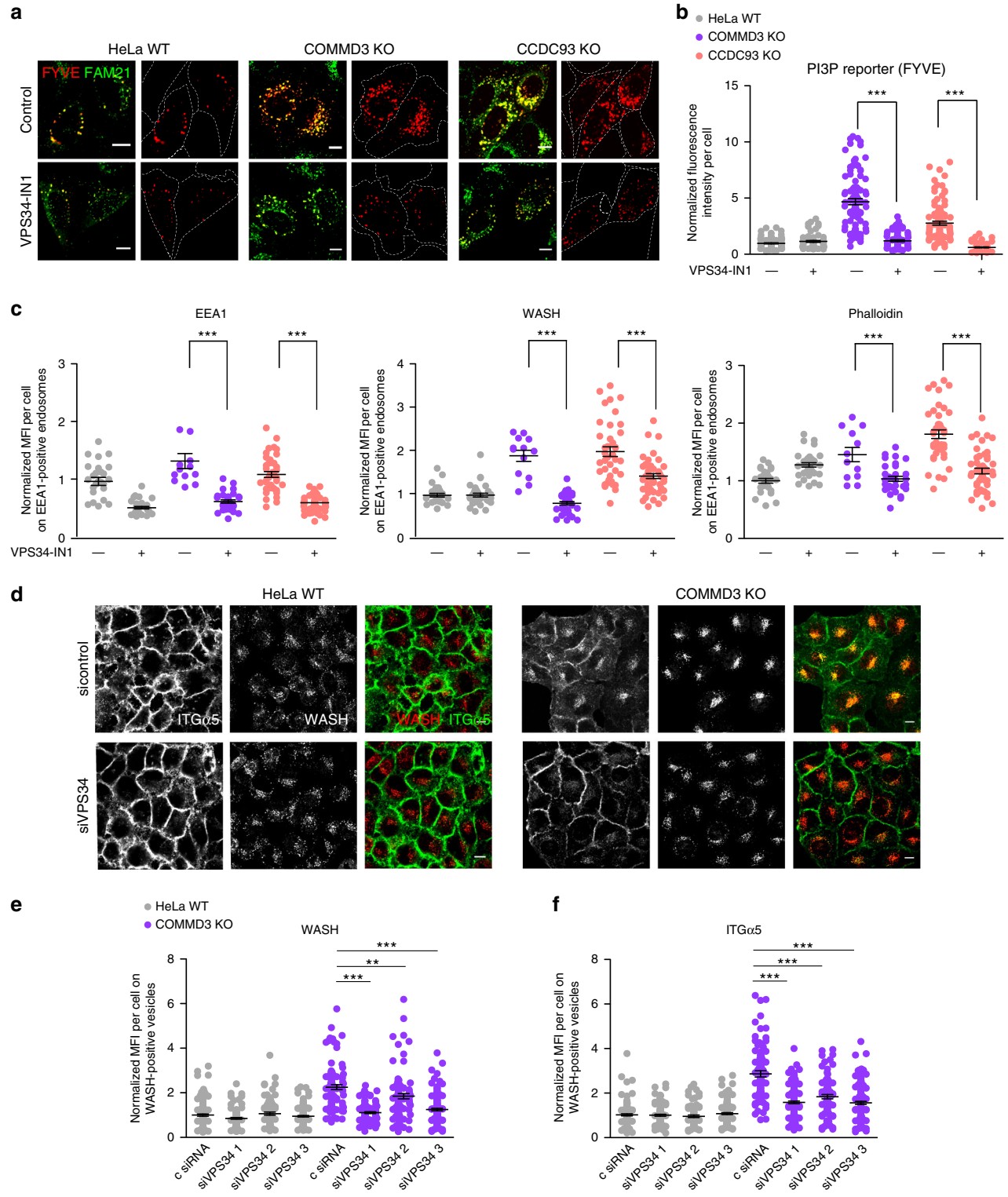

MTMR2 functional defects, specifically its phosphorylation status that controls recruitment to endosomes. It should be noted that inhibition of PIKfyve kinase activity, which prevents PI(3)P conversion to PI(3,5)P$_2$ can also result in increased F-actin on endosomes[53]. However, as PI(3,5)P$_2$ levels were not decreased but rather modestly increased in COMMD3-deficient cells, a defect in PIKfyve activity could not explain the increased PI(3)P levels observed. Hypomorphic MTMR2 mutations are linked to

Charcot–Marie–Tooth syndrome, an inherited form of peripheral neuropathy in humans. However, the spectrum of defects seen with CCC deficiency are far more profound, including neurologic disability and cardiovascular congenital anomalies[67], suggesting that the CCC complex likely mediates other critical events in addition to MTMR2 regulation. This is in line with the multi-subunit architecture of the CCC complex and the evolutionary conservation of all 10 COMMD protein family members. In this

**Fig. 6** VPS34 inhibition restores trafficking in CCC-deficient cells. **a** Confocal images of cells transfected with a PI(3)P reporter (FYVE) and co-stained for FAM21 (left panels). Cells were treated with vehicle (DMSO) or a VPS34 kinase inhibitor (VPS34-IN1). Scale bars, 10 μm. **b** Normalized FYVE fluorescence intensity was quantified. Results for individual cells are plotted, along with the mean and s.e.m. for each group (number of cells: WT = 94, WT VPS34-IN1 = 58, COMMD3 KO = 92, COMMD3 KO VPS34-IN1 = 77, CCDC93 KO = 99, CCDC93 KO VPS34-IN1 = 81); ***$P < 0.0001$ (unpaired two-tailed $t$ test). **c** HeLa WT and two CCC KO cells (COMMD3 and CCDC93 KO) were treated with the VPS34 inhibitor or vehicle control. Quantification of normalized EEA1 mean fluorescence intensity (MFI) per cell (left panel), as well as WASH and phalloidin MFI on EEA1-positive areas (middle and right panels) are shown. Results for individual cells are plotted, along with the mean and s.e.m. for each group (number of cells: WT = 26, WT VPS34-IN1 = 34, COMMD3 KO = 12, COMMD3 KO VPS34-IN1 = 32, CCDC93 KO = 40, CCDC93 KO VPS34-IN1 = 50); ***$P < 0.0001$ (unpaired two-tailed $t$ test against DMSO negative control). **d** Confocal microscopy imaging of ITGα5 and WASH immunofluorescence staining after an ITGα5 recycling experiment. VPS34 silencing (siVPS34) in the indicated cell lines is shown. Representative of 2 independent experiments. Scale bars, 10 μm. **e**, **f** Cells were transfected with control or three independent VPS34 siRNAs and normalized mean fluorescence intensity of WASH per cell **d** or ITGα5 on WASH-positive endosomes **e** was quantified. Results for individual cells are plotted, along with the mean and s.e.m. for each group (number of cells: HeLa control siRNA (csiRNA) = 130, HeLa siVPS34 1 = 113, HeLa siVPS34 2 = 82, HeLa siVPS34 3 = 142, COMMD3 KO csiRNA = 90, COMMD3 KO siVPS34 1 = 120, COMMD3 KO siVPS34 2 = 85, COMMD3 KO siVPS34 3 = 130); ***$P < 0.0001$; **$P < 0.001$ (one-way ANOVA and Dunnett's test to control siRNA)

regard, it is important to highlight that the unique contributions of each COMMD protein to the endosomal sorting process remain poorly understood at the moment. Recent structural studies[68] have shed light into the assembly of COMMD complexes and confirmed the importance of the COMM domain in the dimerization of these proteins[28]. Furthermore, these studies suggest that higher order COMMD hexamers may be possible through additional interactions via their shared helical N-terminal domains. The contribution to the function of the CCC complex that COMMD protein assemblies may provide remains to be evaluated.

In conclusion, we provide evidence that the CCC complex is a negative regulator of WASH activity through its ability to modulate endosomal PI(3)P levels. Future studies aimed at understanding how CCC controls MTMR2 phosphorylation and how PI(3)P impacts WASH complex recruitment and activation, will provide further insight into how deregulation of the CCC complex contributes to cellular physiology and human disease.

## Methods

**Cell culture**. HEK293T and HeLa cell lines were obtained from the American Type Culture Collection (Manassas, VA). Previously reported fibroblast lines used here include embryo fibroblasts from *Commd1* floxed mice[31], embryo fibroblasts from *Wash1* floxed mice reconstituted with stable expression of HA-GFP-WASH[39], and dermal fibroblasts from patients with the *CCDC22* p.T17A mutation[30]. All cell lines were cultured in high-glucose Dulbecco's modified Eagle's medium (DMEM) containing 10% fetal bovine serum and 1% penicillin/streptomycin, in incubators at 37 °C, 5% $CO_2$.

**Plasmids, siRNAs, and transfection methods**. The FLAG-MTMR2 plasmid was a kind gift from Dr. Fred Robinson. The expression vector for MTMR2, pEBB-2xHA-MTMR2, is a derivative of pEBB containing tandem HA tags and the ORF for MTMR2 that was PCR amplified using the FLAG-MTMR2 plasmid as a template. The pEBB-2xHA-MTMR2 S58A mutant construct was generated from pEBB-2xHA-MTMR2 using QuickChange II Site-Directed Mutagenesis Kit (Agilent, Santa Clara, CA) according to the manufacturer's instructions. The dsRed-FYVE PI(3)P reporter plasmid was a kind gift from Dr. David Katzmann (Mayo Clinic). Double-stranded siRNAs for VPS34 were purchased from Thermofisher Scientific with the following references: PIK3C3 (hVPS34) ID s10517, s10518, s10519. MTMR2 siRNA duplexes were obtained from Sigma Aldrich (St. Louis, MO) and included the following: control non-target siRNA (SIC002), human MTMR2 #1 (SASI_Hs01_00132911) and human MTMR2 #2 (SASI_Hs01_00132910). A standard calcium phosphate transfection protocol was used to transfect MTMR2 plasmids in HEK293 cells; the Xtremegene HP transfection reagent (Roche) was used to transfect MTMR2 plasmids in HeLa cells. For the transfection of PI(3)P reporter, Lipofectamine 2000 (Life Technologies) was used. For siRNA experiments, HeLa cells were transfected with control or target siRNA oligonucleotides by using Lipofectamine RNAi MAX (Life Technologies) and cultured for either 48 or 72 h before analysis. For VPS26C and CCDC93 rescue experiments, cells were stably transfected with the lentiviral p-Receiver vector expressing HA-tagged VPS26C or CCDC93. COMMMD3 rescue cells were generated by lentiviral infection of COMMD3 KO cells using pLVX3.FLAG.COMMD3 vector. shRNA-mediated silencing for COMMD1 (mouse or human) and WASH was performed by lentiviral infection using the shRNA expression vector pLKO. TRC. The targeted sequences used are noted in Supplementary Table 1. All

lentivirus experiments were performed with a standard viral vector production and selection protocol[69]. In brief, HEK293T cells were transfected with RSV-rev, Gag/pol, VSV-G, and lentiviral plasmids of interest (pLVX3, pLKO, and pLenti). Forty-eight hours later, medium was collected, filtered, and used to infect target cell lines, which were then subjected to selection in appropriate antibiotics.

**CRISPR/Cas9-mediated gene deletion**. CCDC93 and VPS26C CRISPR/Cas9 knockout cell lines were previously reported[20,27]. COMMD3, VPS35L, and FAM45A knockout cells were generated using CRISPR/Cas9 technology[20]. This was performed through the stable expression of a targeting guide RNA (gRNA) and Cas9, using the pLENTI-CRISPR vector. Clones were isolated through limiting dilution and screened by immunoblot for protein expression. CRIPSR guide RNA sequences used in this study are listed in Supplementary Table 1.

**Antibodies and chemicals**. All antibodies used in this study are listed in Supplementary Table 2. Antibodies to WASH, FAM21, VPS35L, CCDC22, and COMMD proteins were generated and previously reported[27,34,38,40]. Antibodies to FAM45A were generated by Cocalico Biologicals, Inc. (Reamstown, PA) following immunization of rabbits with a keyhole limpet hemocyanin (KLH)-conjugated peptide spanning amino acids 95–110. VPS34-IN1 was purchased from Selleckchem (S7980) and was added to growth media at a final concentration of 1 μM.

**Protein extraction, immunoblotting, and immunoprecipitation**. Whole-cell lysates were prepared by adding Triton X-100 lysis buffer (25 mM HEPES, 100 mM NaCl, 10 mM DTT, 1 mM EDTA, 10% Glycerol, 1% Triton X-100) or MRB buffer (20 mM HEPES pH 7.2, 50 mM potassium acetate, 1 mM EDTA, 200 mM D-Sorbitol, 0.1% Triton X-100) supplemented with protease inhibitors (Roche), as indicated in each experiment. Immunoblotting experiments were performed by resolving the protein samples using 4–12% gradient Novex Bis-Tris gels (Invitrogen), transferred to nitrocellulose membranes and blocked with 5% milk solution in TBS containing 0.05% Tween-20. Subsequently, membranes were incubated with primary antibodies, followed by incubation with HRP-conjugated secondary antibodies[38]. For immunoprecipitation, polyclonal antibody for the protein of interest was added to the cell lysate and incubated at 4 °C. After 1 h, protein G agarose beads (Invitrogen) were added and the samples were incubated for an additional 2 h. For precipitation of HA-tagged proteins, anti-HA affinity matrix (Roche) were added to the cell lysates and incubated at 4 °C for 2–4 h. In all precipitated experiments, beads were washed four times with the lysis buffer and were resuspended in lithium dodecyl sulfate boiling buffer (Invitrogen) and used for immunoblotting as described above. The antibodies used for immunoprecipitation and immunoblotting are detailed in Supplementary Table 2.

**Immunofluorescence staining**. Cells were fixed in ice-cold fixative (4% paraformaldehyde in phosphate-buffered saline; PBS) and incubated for 18 min at room temperature in the dark, followed by permeabilization for 3 min with 0.15% Surfact-Amps X-100 (28314, Thermofisher Scientific, Rockford, IL) in PBS. Samples were then incubated overnight at 4 °C in a humidified chamber with primary antibodies in immunofluorescence (IF) buffer (Tris-buffered saline plus human serum cocktail). After three washes in PBS, the samples were incubated with secondary antibodies (1:500 dilution in IF buffer) for 1 h at room temperature or overnight at 4 °C in a humidified chamber. After four washes in PBS, coverslips were rinsed in water and affixed to slides with SlowFade Anti-fade reagent (Life Technologies, Grand Island, NY). The primary and secondary antibodies used for staining are detailed in Supplementary Table 2. Alexa Fluor 647–phalloidin (A22287), Alexa Fluor 488–phalloidin (A12379), and rhodamine–phalloidin (R415) (Life Technologies, Grand Island, NY) were used to visualize F-actin. Images were obtained using the following confocal microscopes: LSM-710 (×63 /1.4 oil immersion objective lens; Carl Zeiss) or A1R (×60 /1.4 oil immersion objective

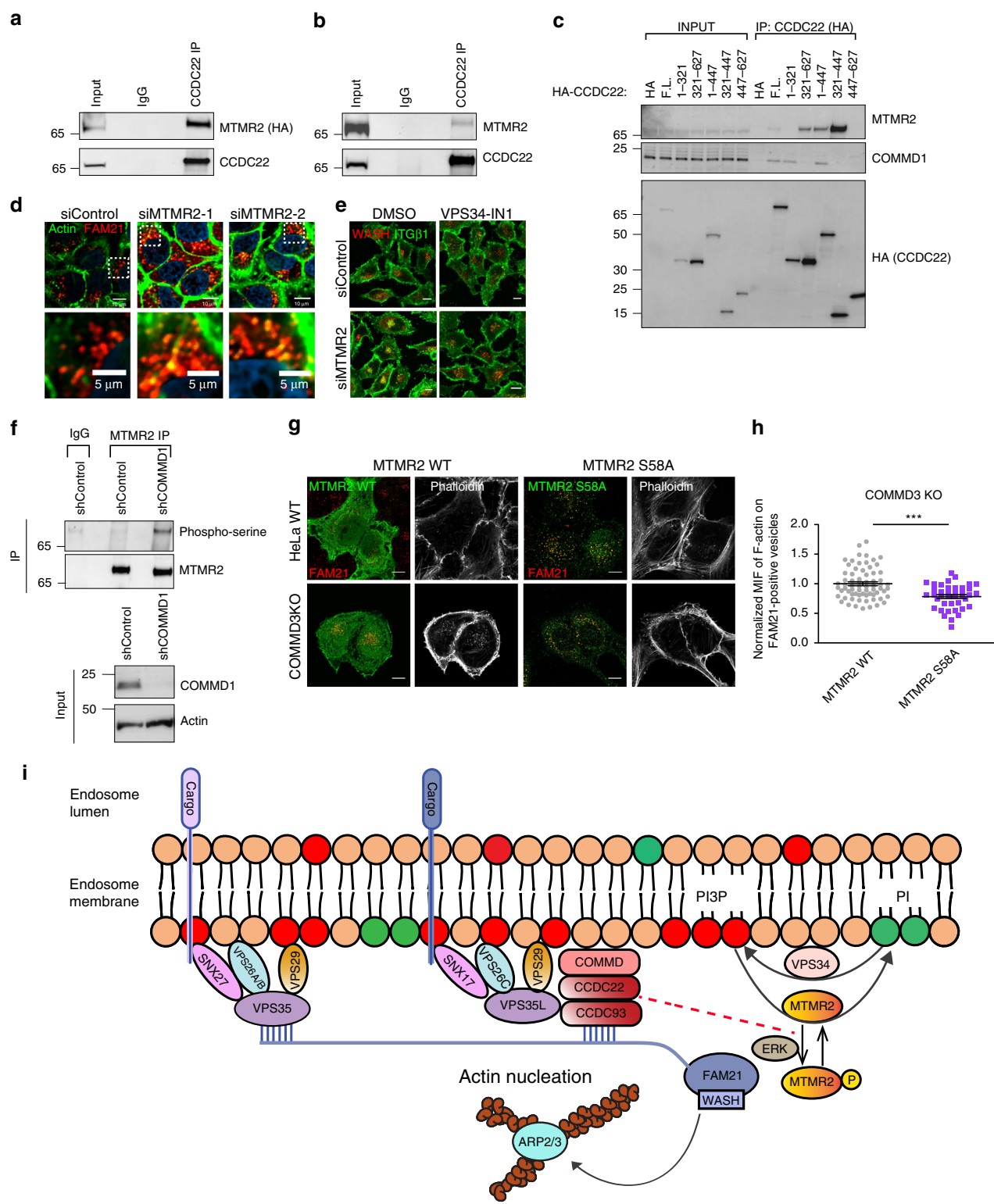

lens; Nikon). Images were acquired using the ZEN software (Carl Zeiss) or the NIS-Elements AR (Nikon) software. Acquisition settings for imaging were identically set for all cell genotypes and treatment groups within each experiment[38].

**Integrin degradation, endocytosis, and recycling**. To assess ITGβ1 lysosomal degradation, cells were incubated with 100 nM Bafilomycin A (Sigma Aldrich, St. Louis, MO) for 24 h, lysed, and protein levels were analyzed by immunoblotting. To assess ITGβ1 endocytosis, cells were dissociated with Cell Dissociation Non-enzymatic Solution (C5914, Sigma Aldrich, St. Louis, MO), and immediately placed on ice. Cell suspension was stained on ice with ITGβ1-biotin antibody for 30 min,

washed three times, and incubated at 37 °C in warm DMEM for the indicated time points. Subsequently samples were placed on ice, washed with cold PBS three times and surface ITGβ1 was labeled with FITC Streptavidin for 30 min (554060, BD Pharmingen). Samples were washed with cold PBS three times and resuspended in FACS buffer (PBS with 2% FBS and 5 mM EDTA). Cells were kept on ice at all times and subjected to FACS analysis. The number of FITC-positive cells was counted and recorded as a percentage of the whole population on BD FACS Canto. Unstained cells and IgG-FITC were used as negative control. Data were analyzed using FlowJo 887. The MFI at the zero time point (ITGβ1 cell surface levels at 4 °C), was set at 100% for each cell line.

**Fig. 7** CCC complex regulates MTMR2 phosphorylation. **a** Immunoprecipitation of CCDC22 from HEK293T cells transfected with pEBB-2xHA-MTMR2. Precipitates were immunoblotted for MTMR2 (HA) and CCDC22. Representative of three independent iterations. **b** Immunoprecipitation of endogenous CCDC22 from HEK293T cell lysates followed by immunoblotting for CCDC22 and MTMR2. Representative of three different iterations. **c** The indicated HA-tagged CCDC22 constructs were transfected in HEK293T cells followed by immunoprecipitation with HA antibody. The recovered material was immunoblotted for endogenous MTMR2 and COMMD1, as well as CCDC22 (HA). Representative of two different iterations. **d** Confocal imaging of F-actin (phalloidin) and FAM21 immunofluorescence staining in HeLa cells transfected with two independent MTMR2 siRNAs or a control oligonucleotide. Data are representative of three independent experiments. Scale bars, 10 μm, 5 μm on zoomed images. **e** Confocal microscopy imaging of ITGβ1 and WASH immunofluorescence staining after an ITGβ1 recycling experiment. Cells were transfected with control or MTMR2 targeting siRNA and treated with VPS34 kinase inhibitor (VPS34-IN1) or vehicle control (DMSO) as indicated. Representative of two independent experiments. Scale bars, 10 μm. **f** Immunoprecipitation of endogenous MTMR2 from cell lysates obtained from HEK293T shControl and shCOMMD1, followed by immunoblotting for phosphoserine, MTMR2 and COMMD1. The experiment is representative of three independent iterations. **g** Confocal imaging for MTMR2 (HA), FAM21, and phalloidin immunofluorescence staining from HeLa WT and COMMD3 KO cells transiently transfected with pEBB-2xHA-MTMR2 and pEBB-2xHA-MTMR2 S58A. Representative images from two separate iterations are shown. Scale bar, 10 μm. **h** Quantification of normalized mean fluorescent intensity (MFI) of F-actin on FAM21-positive vesicles in COMMD3 KO. Results for individual cells are plotted, along with the mean and s.e.m. for each group ($n = 69$ for COMMD3 KO 2HA-MTMR2 WT, $n = 40$ for 2xHA-MTMR2 S58A); ***$P < 0.0001$ (unpaired two-tailed t test). **i** Model depicting how the CCC complex regulates retriever recruitment as well as WASH-dependent actin nucleation on endosomes by regulating PI(3)P levels through MTMR2 phosphorylation/ recruitment. Please refer to the Discussion section for additional details

For the recycling assay, cells were incubated at 37 °C in serum-free DMEM containing 10 μg/mL anti-integrin-β1 or anti-integrin-α5 antibody for 60 min, quickly rinsed with PBS, and then stained for the internalized antibodies by immunofluorescence as described above. To assess the dynamics of the recycling, cells were incubated at 37 °C in serum-free DMEM containing 10 μg/mL anti-integrin-β1 antibody for 30 min, and cell surface antibody was then stripped (PBS, pH 2.5). Subsequently, cells were incubated at 37 °C for 30 min and then fixed and processed for imaging.

**Image quantification**. Fluorescence signal values were quantified using Fiji (ImageJ, NIH). Data were processed with Excel (Microsoft) and plotted with Prism6 (GraphPad). Each dot represents the value from a single cell; the horizontal bar in these graphs represents the mean and the error bars correspond to the standard error of the mean (SEM). In all quantification experiments, the data were normalized by dividing the mean fluorescence intensity (MFI) values of the experimental samples compared with the average mean fluorescence intensity of control cells to derive relative fluorescence intensities (expressed as fold).

Pearson's correlation coefficient was measured using Colocalization Threshold Fiji Plugin within manually outlined regions of interest (ROIs). WASH fluorescence was used to define ROI for Pearson's correlation coefficient between WASH and ITGβ1, EEA1 for Pearson's correlation coefficient between GLUT1 and EEA1, FAM21 for Pearson's correlation coefficient between FAM21 and MTMR2.

For the quantification of actin on WASH-positive vesicles, WASH fluorescence was used to manually define the ROI. The mean fluorescence intensity of WASH within the ROI, and the ROI area was then measured. A similar approach was used for the measurement of cortactin on FAM21-positive vesicles (FAM21 fluorescence was used to define the ROI), ITGα5 and ITGβ1 on WASH-positive vesicles (WASH fluorescence was used to manually define the ROI). For all the above measurements the MFI values were corrected for the ROI area. For the quantification of ARPC2 on VPS35-positive vesicles (VPS35 fluorescence was used to define the ROI), WASH, EEA1, and actin on EEA1-positive vesicles (EEA1 fluorescence was used to define the ROI), actin on FAM21-positive vesicles (FAM21 fluorescence was used to define the ROI), ITGβ1 on WASH-positive vesicles in si-MTMR2-treated cells (WASH fluorescence was used to manually define the ROI), the squares of the same size were set as ROIs, and values were measured within the same area. For the quantification of the FYVE fluorescence reporter, a region covering the entire cell was manually selected (ROI), and FYVE mean fluorescence pixel intensity in the ROI was calculated.

**Quantitative real-time PCR**. Total RNA was extracted from cells using the Trizol method (Life Technologies) according to the manufacturer's protocol. Isolated RNA was subjected to quantitative RT-PCR using the SYBR green and the relative quantification of target transcripts was normalized to the housekeeping gene 18 S ribosomal RNA. The following primer sequences were used for MTMR2; Sense (5′-AAGTATTCCCTGAAAATGGGTGG-3′) and Antisense (5′-AAACTGGGAT ACGGCCTCTTG-3′).

**Blue native electrophoresis and immunoblotting**. Cell lysates were prepared using NativePAGE Sample Prep kit (ThermoFisher Scientific). In brief, the cells were lysed on ice for 10 mins in 1 × sample buffer containing 1% Digitonin and Halt protease and phosphatase inhibitor cocktail followed by spin at 20,000×g for 30 mins. Protein concentration was determined in the lysate using Biorad protein assay reagent. According to manufacturer's instructions, equal amounts of protein were loaded to NativePAGE 3–12% Bis-Tris protein gels, with one lane containing NativeMark Unstained protein standard and transblotted to PVDF membranes. After transfer, the proteins were fixed by incubating the membrane in 8% acetic acid for 15 mins, followed by immunoblotting as described above.

**Protein affinity purification and proteomics**. CCDC93 and VPS26C knockout HeLa cells generated by CRISPR/Cas9 as detailed above were reconstituted with either HA empty vector (control sample) or HA-tagged CCDC93 and VPS26C, respectively (experimental samples), using the lentivirus system and subjected to negative selection. HA-, HA-CCDC93-, and HA-VPS26C-transduced knockout cells were grown in large batches and lysed in Triton-X lysis buffer. Cell lysates were cleared and equal amounts of protein were then added to HA-resin to capture HA-tagged proteins. HA beads were washed using high stringency buffer (Triton-X lysis buffer supplemented to a total of 350 mM NaCl) and proteins were eluted using HA peptide (1 mg/mL). Trichloroacetic acid precipitation was performed on the eluted material and the resulting pellet was resuspended in 3 × LDS sample buffer with DTT. Short electrophoresis was performed to allow proteins to enter the upper portion of the gel, followed by coomassie gel staining and excision of gel slices from each sample (control, HA-CCDC93, HA-VPS26C, $n = 1$ for each group). Protein gel pieces were reduced and alkylated with DTT (20 mM) and iodoacetamide (27.5 mM). A 0.1 μg/μL solution of trypsin in 50 mM triethy-lammonium bicarbonate (TEAB) was added to completely cover the gel, allowed to sit on ice, and then 50 μL of 50 mM TEAB was added and the gel pieces were digested overnight (Pierce). Following solid-phase extraction cleanup with an Oasis MCX μelution plate (Waters), the resulting peptides were reconstituted in 10 μL of 2% (v/v) acetonitrile (ACN) and 0.1% trifluoroacetic acid in water: 2 μL were injected onto an Orbitrap Fusion Lumos mass spectrometer (Thermo Electron) coupled to an Ultimate 3000 RSLC-Nano liquid chromatography system (Dionex). Samples were injected onto a 75 μm i.d., 50-cm long EasySpray column (Thermo), and eluted with a gradient from 0 to 28% buffer B over 60 mins. Buffer A contained 2% (v/v) ACN and 0.1% formic acid in water, and buffer B contained 80% (v/v) ACN, 10% (v/v) trifluoroethanol, and 0.1% formic acid in water. The mass spectrometer operated in positive ion mode with a source voltage of 2.4 kV and an ion transfer tube temperature of 275 °C. MS scans were acquired at 120,000 resolution in the Orbitrap and up to 10 MS/MS spectra were obtained in the ion trap for each full spectrum acquired using higher-energy collisional dissociation for ions with charges 2–7. Dynamic exclusion was set for 25 s after an ion was selected for fragmentation.

Raw MS data files were converted to a peak list format and analyzed using the central proteomics facilities pipeline (CPFP), version 2.0.3. Peptide identification was performed using the X!Tandem and open MS search algorithm (OMSSA) search engines against the human protein database from Uniprot (Release August 2014), with common contaminants and reversed decoy sequences appended. Fragment and precursor tolerances of 20 ppm and 0.5 Da were specified, and three missed cleavages of trypsin (cleavage after K or R except KP/RP) were allowed. Carbamidomethylation of Cys was set as a fixed modification, with oxidation of Met set as a variable modification. An additional requirement of two unique peptide sequences per protein was used for protein identification. Both protein and peptide FDR were set to 1%. Using the spectral index as a semi-quantitative indication of protein abundance, we calculated enrichment ratios as the ration of any given protein in the experimental samples (HA-CCDC93 and HA-VPS26C) compared with the empty vector control. As a cutoff, proteins with a minimum enrichment of 10-fold and whose spectral indexes were within one order of magnitude of the spectral index of the bait were selected as putative interacting proteins.

**Protein interactome database analysis**. Protein baits were defined as those identified as common partners of CCDC93 and VPS26C that had been experimentally defined by mass spectrometry (Venn diagram shown in Fig. 1a). For each of these factors, we tabulated all reported interacting partners in Bioplex 2.0[46]. In addition, we used Homologene and Blast to identify potential homologs in *Drosophila*. Interacting partners in *Drosophila melanogaster* were identified in NCBI Gene, which collates data from three prior interactome studies[47–49]. For each reported interacting partner, mammalian homologs of *Drosophila* genes were again defined as above (Homologene and Blast). Finally, the number of times that any interacting partner was identified was tabulated and normalized by the maximum theoretical occurrences in the examined gene matrix, which is presented in the form of percentage.

**Pyrene-actin assembly assays**. WASH knockout mouse embryonic fibroblasts reconstituted with HA-GFP-WASH or HEK293T cells were stably transfected with shControl or shCommd1 lentiviral expression vectors, targeting mouse of human COMMD1, respectively. In vitro actin assembly on beads using cell lysates was performed. Cells were lysed in NP-40 lysis buffer (50 mM Tris Cl pH 7.7, 150 mM NaCl, 0.5% NP-40, 2 mM $MgCl_2$ and 1 × protease inhibitor cocktail). Lysate protein concentrations were measured and equal amounts of protein was used to immunoprecipitate the WASH complex using HA antibodies in the case of murine cells expressing HA-GFP-WASH, or FAM21 antibodies in the case of HEK293T cells. The antibodies were precipitated by protein G beads, which were washed with NP-40 lysis buffer containing 2 mM $MgCl_2$. Actin assembly assays were set up by incubating beads with 4 μM actin and 10 nM Arp2/3 complex in KMEI buffer (15% (w/v) glycerol, 50 mM KCl, 1 mM $MgCl_2$, 1 mM EGTA, and 10 mM imidazole; pH 7.0) for 1 h at room temperature. Further, beads were fixed in 3% paraformaldehyde and F-actin on beads was stained with Alexa 568-conjugated phalloidin and imaged. Quantification of fluorescence images was performed using Fiji.

**Measurement of phosphorylated PI levels**. PI measurements by HPLC were performed[70]. In brief, cells were grown to 60–70% confluence, washed with PBS and incubated with inositol-labeling medium (containing custom-made inositol-free DMEM (11964092; Life Technologies), 10 μCi/mL of myo-3H-inositol (GE Healthcare), 10% dialyzed FBS (26400; Life Technologies), 20 mM Hepes, pH 7.2–7.4, 5 μg per mL transferrin (0030124SA; Invitrogen), and 5 μg/mL insulin (12585–014; Invitrogen) for 48 h. After 48 h, cultures were rinsed with PBS and treated with 1 mL 4.5% (v/v) perchloric acid for 15 min at room temperature; plates were gently rotated every 2 mins to prevent cells from drying. Cells were scraped off with a Cell lifter (3008, Costar) and spun at 12,000 × g for l O min at 4 °C. Pellets were washed at room temperature with 1 mL, 0.1 M EDTA and resuspended in 50 μL water. To deacylate lipids, samples were transferred to a glass vial, mixed with 1 mL of methanol, 40% methylamine/n-butanol (45.7% methanol, 10.7% methylamine, 11.4% n-butanol) and incubated at 55 °C for 1 h. Samples were vacuum dried, resuspended in 0.5 mL water, extracted twice with an equal volume of butanol/ethyl ether/ethyl formate (20:4: 1, v/v). The aqueous phase was vacuum dried, resuspended in 70 μL water and 50 μL of each sample was analyzed by HPLC (Shimadzu UFLC CBM-20A Lite) using an anion exchange 4.6 × 250 mm column (Cat.# 4621-1505, PartiSphere SAX, Whatman). A gradient of 1 M (NH4)2HPO4, pH 3.8 (pH adjusted with phosphoric acid) was used at 1 mL/min flow rate: 0% 5 min; 0–2% 15 min; 2% 80 min; 2–10% 20 min; 10% 65 min; 10–80% 40 min; 80% 20 min; 80- 0% 5 min. Radiolabelled eluate was detected by an inline flow scintillation analyzer (Beta-RAM model 5—RHPLC Detector, LabLogic). A 1:2 proportion of eluate to scintillant (Flow Logic HS, LabLogic) was used with a 3 mL/min flow rate. Fractions were analyzed by bunching counts every 6 sec. The data were collected and analyzed by Laura-4 software (LabLogic).

For comparison of phosphatidylinositol polyphosphate levels, the raw counts in each peak was expressed as a percentage of total phosphatidylinositol, calculated from summation of the counts of the detectable glycerophosphoinositol phosphate peaks (Pl, PI3P, PI4P, PI5P, Pl(3,5)P2, PI(3,4,5)P3, and PI(4,5)P2).

**VPS34 in vitro kinase activity assay**. In vitro kinase assays were performed following the manufacturer's instructions (Echelon Biosciences Inc, Salt Lake City, UT). In brief, HeLa cells were grown to 70% confluency and lysed with ice-cold lysis buffer (20 mM Tris, pH 7.5, 137 mM NaCl, 1 mM $MgCl_2$, 1 mM $CaCl_2$, 1% NP-40, 10% glycerol), supplemented with protease inhibitor cocktail. Lysate protein concentrations were measured, and equal amounts of protein were used to immunoprecipitate anti-VPS34 (Echelon, 1:125) or anti-rabbit IgG overnight at 4 °C on a rotating wheel, followed by a 1-h incubation with the washed protein A sepharose beads, while rotating at 4 °C. Immune complexes were washed with freshly prepared buffers in following order: three times with PBS with 1% NP-40, three times with 100 mM Tris-HCl pH 7.5, 500 mM LiCl, two times with TNE (10 mM Tris-HCl, pH 7.5, 100 mM NaCl, and 1 mM EDTA), two times with VPS34 kinase reaction buffer (10 mM Tris pH 8, 100 mM NaCl, 1 mM EDTA, 10 mM $MnCl_2$, and 50 μM ATP). PI(3)P production was detected using an ELISA Class III PI3-Kinase Kit K-3000 (Echelon Biosciences Inc, Salt Lake City, UT), following the manufacturer's instructions. To control for protein recovery, samples were eluted in 4 × sodium dodecyl sulfate (SDS) sample buffer by boiling for 5 min, and subsequently loaded onto SDS–PAGE gels for immunoblotting, as described above.

**Study approval**. All patient-related studies were performed with the written consent of the participants or their legal guardians, after review and approval of the study protocols by the human research ethics committee of the Women's and Children's Health Network (Adelaide, South Australia, Australia). This protocol authorized the derivation of dermal fibroblasts from patients with *CCDC22* mutations. Murine studies were approved by the UT Southwestern Institutional Animal Care and Use Committee under study number APN 102011 and by the Mayo Clinic Institutional Animal Care and Use Committee under study number A21314. These protocols authorized the preparation of mouse embryo fibroblast cell lines utilized in the studies presented here.

**Statistical analysis**. For all quantitative data, mean ± SEM is shown. Graph preparation and statistical analyses were performed using Prism 6 (GraphPad) or Excel (Microsoft). One-way analysis of variance and Dunnett's test or unpaired Student's two-tailed $t$ test was used to test for statistical significance. For all experiments, a $P$ value of < 0.05 was considered statistically significant.

**Reporting summary**. Further information on research design is available in the Nature Research Reporting Summary linked to this article.

## Data availability
All uncropped immunoblots associated with Figs. 1e, 2b, 4e, 7a, 7b, 7c, 7f, and Supplementary Figs. 1a, b, d, 2c, 4d, 6d, 7a, c, and 8a are shown in Supplementary Fig. 9. The source data underlying Figs. 2c, 3b, 3d, 3f, 4a, 4d, 4f, 5b, 5c, 5e, 5f, 6a, 6b, 6c, 6e, 6f, 7g, and Supplementary Figs. 2a, d, e, 3a, b, 4a, c, 5b, 6e, 7b, 8b, d and e are provided as a Source Data file. Proteomic data associated with Fig. 1a are available from the ProteomeXchange consortium via the MassIVE partner repository with the accession number MSV000084217 [https://massive.ucsd.edu/ProteoSAFe/dataset.jsp?accession=MSV000084217]. Additional data that support the findings of this study are available from the corresponding author upon reasonable request.

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

## Acknowledgements

We thank members of our laboratories for critical discussions. This research is supported by the following NIH grants: R01DK073639 (E.B.), R01DK107733 (D.D.B. and E.B.), R01NS064015 (L.S.W.), R01NS099340 (L.S.W.), and K01DK106346 (A.S.). Additional support included the Natural Science Foundation of China (NSFC) grant #31671477 (D.J.). D.J. is a One Thousand Talents program scholar.

## Author contributions

Initial conceptualization: D.D.B. and E.B. Evolution of conceptualization: A.S., A.F., E.B., and D.D.B. Investigation, A.S., A.F., and S.S.P.G. Writing—original draft: A.S., A.F., E.B., and D.D.B. Writing—review and editing, all authors. Funding acquisition: A.S., L.S.W., D.J., E.B., and D.D.B. Resources: B.L.O., A.L., D.J., J. S., and K.-H.H. Supervision, D.D.B., E.B., and L.S.W.

## Additional information

**Competing interests:** The authors declare no competing interests.

**Peer Review Information** *Nature Communications* thanks Kwang Pyo Kim and other, anonymous, reviewer(s) for their contribution to the peer review of this work. Peer reviewer reports are available.

