## [Peer Review File · Nature Communications]

Reviewers' comments:

Reviewer #1 (Remarks to the Author):

This manuscript is a follow up study from the authors previous identification of the CCC complex and retriever as complexes required for the endosomal sorting of cargo. The authors identify that deletion of components of the CCC complex or VPS35L (a retriever component) leads to defects in gross cargo sorting. The authors go on to suggest that this is due to an increase in PI3P levels on the endosome due to a defect in recruiting the PI3P phosphatase MTMR2, leading to excessive WASH recruitment and WASH-dependent actin polymerisation on the endosomal membrane.

Main points:

Whilst parts of this manuscript are novel (centred around an increased WASH localisation and correlation with PI3P levels), a significant amount of the conclusions have already been published by the authors in their recent collaborative paper describing the identification and functional validation of retriever. There is limited mechanistic evidence to say that it is the CCC complex that interacts with and recruits MTMR2 to the endosomal membrane. The authors show that FAM21 and the CCC complex associate with a small amount of MTMR2 (fine as it is an enzyme) but no further biochemistry is shown to probe the mechanism of phosphatase binding. The authors propose that the CCC complex recruits MTMR2 to endosomes but the imaging demonstrating this is not convincing – the construct used does not appear to localise to endosomes, which contrasts to a FLAG tagged construct used in the cited Franklin et al., study. As the authors note, no mechanism for how PI3P affects WASH recruitment is provided. Overall, the manuscript provides a series of correlated observations but does not provide a 'detailed mechanistic analysis'.

Specific points:

p7, line 142. There is a lack of clarity relating to some of the data shown in the present study with that recently published by the authors as part of a collaborative identification of the retriever (and see below). Retriever has been reconstituted as a heterotrimeric complex and hence it can assemble in the absence of the CCC complex. This should be more clearly stated at this point so as to place the data shown in Figure 1 into the correct context.

Figure 1A: This figure is not particularly convincing. The blots do not line up which makes interpretation of the data difficult. Whilst the authors state that the WASH complex is eluting in a peak distinct to retriever and retromer, FAM21 at least is eluting with retriever and the CCC complex. Moreover, VPS26C seems to co-elute with retromer and parts of retromer co-elute with

retriever. Perhaps collecting smaller fractions (it is not noted what size the fractions are) from the SEC column may aid in distinguishing the complexes?

Figure 1D: Have the cropped blots originated from the same or from separate experiments? (the VPS29 blot appears to be uncropped). In addition, while VPS35L seems to have been enriched in the VPS35L IP, the same cannot be said for the other IP's. For example, the CCDC93 IP does not appear to have been enriched for CCDC93 especially when you compare with the amount of CCDC93 IP'd by other antibodies.

P7 line 143. In their retriever paper the authors provided clear and quantified data to establish that the CCC complex links retriever to WASH-containing endosomes. The final sentence of this section (p8, line 154-156) therefore serves to confirm this finding and is not by itself a new conclusion.

In Figure 2A, how do the authors reconcile that there is less VPS35L interacting with FAM21 under KO conditions when, in each case, there is less VPS35L expressed in the KO cell lines than the wild type control. Has this been accounted for or is the reduced association of VPS35L with FAM21 simply due to lower levels of VPS35L to begin with?

p. 8, line 159-160. The authors have previously published (experiments that they repeat in the present manuscript and show as Figure 2C and 2D) that for integrin recycling there is a clear functional link between retriever and the CCC complex in the WASH-dependent sorting of this adhesion molecule (the authors even refer to this in their introduction (p. 4, lines 71-75)). 'Not suggested by prior reports' is therefore a confusing statement.

Do the authors have any thoughts as to why loss of FAM45A function gives a GLUT1 phenotype but not an integrin phenotype?

Can the authors confirm that the MFI takes into account the area of the ROI? This is especially important as the swelling and collapse of the endosomal system may affect quantification if the area is not included.

Does the deletion of the CCC complex have any effect on the entire WASH complex assembly? The formation of the complex is thought to regulate WASH activity and hence this is an important consideration. The authors have showed that FAM21 still associates with WASH (Supplementary Figure 3), but what about the other components of the WASH complex? This extends to Figure 4C. Is the GFP-WASH assembling into the full WASH complex when IP'ed from COMMD1 suppressed cells

or is the increased F-actin polymerisation due to a de-stabilised WASH complex that has reduced inhibitory activity?

Figure 6B: Given the suggestion that increased PI3P increases recruitment of WASH to endosomes, is it not surprising that inhibition of PI3P synthesis does not have a negative effect on WASH localisation in wild type cells?

Figure 6D: Is the affect of the VPS34 siRNA in the COMMD3 KO cells statistically different to that in HeLa wild type cells? If so, is this a partial rescue of the phenotype? Does the collapsing of endosomes also rescue?

Figure 7A: Is this blot labelled correctly? Should the CCDC22 IP not be labelled as the FAM21 IP (and vice versa)?

In Supplementary Figure 7C are data derived from the same experiment?

Figure 7G: The imaging here does not look like the imaging in the paper the authors have referenced where the construct is either cytosolic or on Rab5 vesicles under serum starvation (no reticular phenotype is seen). Is the GFP tag on this construct affecting the localisation of the enzyme, and if so, can it be used in these specific experiments to draw any firm conclusion?

Reviewer #2 (Remarks to the Author):

This is a highly interesting manuscript that describes novel aspects of the interaction of the endosomal transmembrane protein cargo sorting complexes CCC, Retriever, Retromer and WASH and identifies the CC complex as a negative regulator of the WASH complex through regulation of MTMR2. Loss of CCC is shown to lead to an endosomal accumulation of actin polymerisation capable WASH complex. The MTMR2 mediated PI feedback loop is an important discovery, especially in the light of recent work that showed endosomal WASH retention is dependent on HRS, itself a PI3P binding protein (MacDonald et al. JCB2018).

It is interesting to see that an apparent overactivation of the WASH complex can stop recycling without significantly affecting retriever or retromer levels on the endosome. The authors need to

point this out more clearly. While the possible conclusions are exciting, technical points need to be strengthened.

In detail:

- The CCC-retriever-retromer interactions have only recently been described and seem to be very complex, this is a very important aspect of this paper and the authors should quantify all their knock-out cell line complex protein blots in Fig.1 to add valuable information for the readership.

-there are numerous experiments (for example Figs. 2c,d,e,f; 5C) where a factor whose endosomal association is changed in the knockout cells was used to quantify the association of other endosomal factors (for example WASH, EEA1). Can the authors please use endosomal protein factors that show unchanged association in their knockout cell lines or just use dextran loaded endosomes to quantify endosomal association of their proteins of interest.

Line number:

103 Figure1a: This gel filtration experiment is very interesting. However, the results for the size of the WASH are somewhat at odds with previously published blue native (BN) gels results (Tyrrell et al. Pigment Cell Research) and do not match the molecular weight of a pentameric complex. It would be good if the authors could perform BN gel electrophoresis to reconcile their data with the published results. In addition, this would give a better idea about CCC and retromer complexes.

Figure1D can the authors explain why only VPS29 is one connected blot and the other proteins are not? Also, the VPS26C bands seem to be at an angle? Please include the original uncut blots as supplementary data to make it possible to interpret this experiment. In addition all blots in Fig.1 need MW markers.

165: ITGbeta1 recycling is also WASH dependent (Duleh&Welch Cytoskeleton 2012)

The assay in Fig. 2C does not necessarily show recycling of proteins. the intracellular accumulation could be caused by a.) increased endocytosis b.) defective degradation c.) decreased recycling. To make a statement about recycling all three possibilities need to be experimentally tested. Established biochemical assays should be used.

300 if starvation results in increased MTMR2 activity it should result in reduced WASH and actin endosomes. The authors need to test this experimentally or remove the passage.

Reviewer #3 (Remarks to the Author):

Professors Billadeau and Burstein report CCC complex controls membrane protein recycling by regulating PI(3)P levels within the endosomal compartment. The subject is interesting and timely, and the work is well done. This study is a well-presented paper with good assumptions and proof of it. But it is doubtful whether it is appropriate to publish in a prestigious journal such as Nature Communications as a study of the mechanisms of already known phenomena, rather than a study of the discovery of new biological processes.

Billadeau and coworkers have earlier reported the discovery of an ancient and conserved multi-protein complex which orchestrates cargo retrieval and recycling (ref. Nat Cell Biol. 2017 Oct; 19(10): 1214–1225). They proposed the the importance of SNX17 in retromer-independent cargo sorting pathway. In continuation of this work, in the present manuscript the authors uncover the critical role of CCC complex in endosomal trafficking of cargo proteins and report that endosomal PI(3)P regulation by the CCC complex controls membrane protein recycling.

Overall, it is an interesting and useful piece of work. The work is sound and has been carried out with great care taking care of all the possible questions. However, my major concern is, in the light of their earlier paper in Nat Cell Biol. 2017 whether the novelty level of this paper is good enough for Nature communications. The following points should be addressed for publication.

1. As in this study, studies that analyze protein interactions using LC-MS after IP often have problems associated with identification of false positive proteins. The authors should elaborate in detail how false positive proteins were removed from the list of identified proteins.

2. Authors performed mass spectrometric identification of interacting proteins with CCC complex and retriever complex following HA-CDC93 or HA-VPS26C immunoprecipitation. In order to further clarify these results, it is recommended to perform and compare the same immunoprecipitation separation and mass spectrometry using another different CCC complex component and retriever complex component.

3. The increase of the PI(3)P reporter signal (~ 2.5 fold) in endosomes from COMMD3 and CCDC93 KO cells, (Figure 5d, e) is the most important finding in this study as the authors highlighted it in the

title. However, Figure 5e shows that there is a possibility that the PI(3)P concentration is interpreted as an overall increase due to the large deviation of the upward in COMMD3 KO. Statistically, a detailed description should be added to see if this deviation has been handled properly.

We thank the reviewers for the time and care spent evaluating our manuscript. The reviews included a number of excellent suggestions for experiments that we have done to provide additional mechanistic insight into how loss of CCC complex results in increased PI(3)P on endosomes. To facilitate the review of this revised manuscript, please note that we have highlighted in gray any substantial additions to the text.

Reviewer #1 (Remarks to the Author):

This manuscript is a follow up study from the authors previous identification of the CCC complex and retriever as complexes required for the endosomal sorting of cargo. The authors identify that deletion of components of the CCC complex or VPS35L (a retriever component) leads to defects in gross cargo sorting. The authors go on to suggest that this is due to an increase in PI3P levels on the endosome due to a defect in recruiting the PI3P phosphatase MTMR2, leading to excessive WASH recruitment and WASH-dependent actin polymerisation on the endosomal membrane.

Main points:

1) Whilst parts of this manuscript are novel (centred around an increased WASH localisation and correlation with PI3P levels), a significant amount of the conclusions have already been published by the authors in their recent collaborative paper describing the identification and functional validation of retriever.

Response: The initial data provided in the paper has been largely reformatted in this revision to address questions of redundancy with our prior work. The main point that the data is trying to convey is that the CCC complex is not simply a molecular assembly that extends retriever function but that it has intrinsic and broader functions in cargo recycling. From that foundation, we proceed to demonstrate for the first time the mechanism by which the CCC complex regulates PI3P on endosomal membranes, how this impact WASH complex recruitment and activity, and how this affects cargo recycling. All these insights are new advances from everything previously known or published in this field.

2) There is limited mechanistic evidence to say that it is the CCC complex that interacts with and recruits MTMR2 to the endosomal membrane. The authors show that FAM21 and the CCC complex associate with a small amount of MTMR2 (fine as it is an enzyme) but no further biochemistry is shown to probe the mechanism of phosphatase binding.

Response: We are grateful for this insightful comment. In this revision we have investigated this point a great deal and we are pleased that this has led us in a very important direction. First, we provide domain-mapping analysis that demonstrates that the binding between MTMR2 and CCDC22 can be mapped to a region of CCDC22 that by itself does not associate with the rest of CCC complex, indicating that it is likely that CCDC22 may be bringing MTMR2 into the complex. Second, we examined the endosomal recruitment of MTMR2 to endosomes. As others have previously reported, we find that MTMR2 phosphorylation prevents endosomal recruitment. Importantly, as the reviewer astutely noted, the amounts of protein on the endosome are ordinarily very low and difficult to detect, likely because as an enzyme low levels of recruitment are all that is required under physiologic conditions. Third, we show that MTMR2 phosphorylation is highly increased in CCC-deficient cells, thus resulting in reduced endosomal recruitment and explaining the PI3P phenotype seen. And finally, we show that bringing

MTMR2 to endosomes in CCC deficient cells (by bypassing the phosphorylation-dependent regulation of MTMR2 recruitment) leads to resolution of the excessive actin accumulation on endomembranes.

3) The authors propose that the CCC complex recruits MTMR2 to endosomes but the imaging demonstrating this is not convincing – the construct used does not appear to localise to endosomes, which contrasts to a FLAG tagged construct used in the cited Franklin et al., study. As the authors note, no mechanism for how PI3P affects WASH recruitment is provided.

Response: As suggested, we have tested several constructs of MTMR2 to study its cellular localization, including FLAG and HA constructs, and we have removed the data utilizing longer tags such as AcGFP. In agreement with prior work (Franklin et al., 2013. JCS 126:1333-1344), none of these constructs show substantial endosomal localization under baseline conditions, unless we utilize a point mutant form of MTMR2 that is resistant to phosphorylation (Figure 7g). Nonetheless, we show quite striking effects of the CCC complex on the phosphorylation status of MTMR2 as pointed out above (Figure 7f). As far as the second point made here concerning PI3P-mediated recruitment of WASH, a detailed analysis of that question is best suited for a separate study since the focus of this paper is on the activities of the CCC complex itself.

4) Overall, the manuscript provides a series of correlated observations but does not provide a ‘detailed mechanistic analysis’.

Response: We respectfully disagree with this characterization of the paper, particularly in its present form. As outlined above, we demonstrate that the CCC complex regulates MTMR2 phosphorylation, a key mechanism for MTMR2 endosomal recruitment, and go on to show that trafficking is highly dependent on MTMR2-mediated PI3P to PI conversion. Absent this step, there is exaggerated WASH recruitment, supraphysiologic deposition of actin on endosomal sorting domains, and endosomal retention of cargo proteins. We believe that the degree of mechanistic insight provided by this work is indeed worth publication in Nature Communications.

Specific points:

5) p7, line 142. There is a lack of clarity relating to some of the data shown in the present study with that recently published by the authors as part of a collaborative identification of the retriever (and see below). Retriever has been reconstituted as a heterotrimeric complex and hence it can assemble in the absence of the CCC complex. This should be more clearly stated at this point so as to place the data shown in Figure 1 into the correct context.

Response: We have made note of this comment and amended the text to be more clear about it. Moreover, thanks to other suggestions in this review (specifically, to examine the mass of these complexes using blue native gels), we show here that retriever is indeed seen as a complex of about 240 kDa in vivo (similar mobility to retromer), with CCC being a distinct 700 kDa complex (Figure 1b). In addition, we are also able to show that VPS35L is present in both the retriever complex as well as in CCC (Figure 1c,d), in agreement with the functional data that indicates that VPS35L behaves as other CCC subunits as far as actin hyper-accumulation.

6) Figure 1A: This figure is not particularly convincing. The blots do not line up which makes

interpretation of the data difficult. Whilst the authors state that the WASH complex is eluting in a peak distinct to retriever and retromer, FAM21 at least is eluting with retriever and the CCC complex. Moreover, VPS26C seems to co-elute with retromer and parts of retromer co-elute with retriever. Perhaps collecting smaller fractions (it is not noted what size the fractions are) from the SEC column may aid in distinguishing the complexes?

Response: Taking into consideration this and other comments, we have removed this data altogether since the resolution between 2000 and 630 kDa is poor by this method. Instead, we decided to utilize blue native gels as suggested (see Figure 1b-d). By this approach, we can show distinct retriever and CCC complex peaks (additional details noted above), and have better resolution of the mobility of WASH and retromer.

7) Figure 1D: Have the cropped blots originated from the same or from separate experiments? (the VPS29 blot appears to be uncropped). In addition, while VPS35L seems to have been enriched in the VPS35L IP, the same cannot be said for the other IP's. For example, the CCDC93 IP does not appear to have been enriched for CCDC93 especially when you compare with the amount of CCDC93 IP'd by other antibodies.

Response: All lanes were run together and the experiment was presented this way only to allow for different exposures to be presented in one consolidated panel, now shown in Figure S1d. To avoid any confusion, all inputs are shown as a separate panel, but figure S9 contains all the uncropped blots for further review.

8) P7 line 143. In their retriever paper the authors provided clear and quantified data to establish that the CCC complex links retriever to WASH-containing endosomes. The final sentence of this section (p8, line 154-156) therefore serves to confirm this finding and is not by itself a new conclusion.

Response: We have made that adjustment as suggested; please note that the data presented here is more rigorous as it relies on two distinct CRISPR knockout cell lines, and we believe that this is still quite valuable.

9) In Figure 2A, how do the authors reconcile that there is less VPS35L interacting with FAM21 under KO conditions when, in each case, there is less VPS35L expressed in the KO cell lines than the wild type control. Has this been accounted for or is the reduced association of VPS35L with FAM21 simply due to lower levels of VPS35L to begin with?

Response: The experiment has been repeated using two isogenic cells lines which are much more closely matched to each other: COMMD3 KO cells reconstituted to reexpress FLAG-COMMD3 or transduced with an empty vector virus. In this setting, the expression of VPS35L is only minimally different. When we IP VPS35L from both lines, and with equal IP recovery of VPS35L, we see no binding of the WASH complex to VPS35L in the COMMD3 deficient line (see Fig 2b), confirming the original observation.

10) p. 8, line 159-160. The authors have previously published (experiments that they repeat in the present manuscript and show as Figure 2C and 2D) that for integrin recycling there is a clear functional link between retriever and the CCC complex in the WASH-dependent sorting of this adhesion molecule (the authors even refer to this in their introduction (p. 4, lines 71-75)). 'Not suggested by prior reports' is therefore a confusing statement.

Response: We edited this text as follows:

“If the sole function of the CCC complex is to link retriever to endosomes, it would be predicted that its deficiency should phenocopy retriever loss, which **is in disagreement with findings from prior reports.**”

We hope that this statement addresses the concerns of clarity that were pointed out.

11) Do the authors have any thoughts as to why loss of FAM45A function gives a GLUT1 phenotype but not an integrin phenotype?

Response: This is an interesting question worthy of additional investigation. At this stage, it appears that FAM45A is more important in the regulation of retromer rather than retriever cargos. This is included in the discussion. Further investigation of this question will require future studies.

12) Can the authors confirm that the MFI takes into account the area of the ROI? This is especially important as the swelling and collapse of the endosomal system may affect quantification if the area is not included.

Response: We thank the reviewer for this important comment. We have gone back and corrected all the MFI data obtained using manual outlined ROIs for the area size, except for the MFI data obtained in squares of the same size.

13) Does the deletion of the CCC complex have any effect on the entire WASH complex assembly? The formation of the complex is thought to regulate WASH activity and hence this is an important consideration. The authors have showed that FAM21 still associates with WASH (Supplementary Figure 3), but what about the other components of the WASH complex? This extends to Figure 4C. Is the GFP-WASH assembling into the full WASH complex when IP'ed from COMMD1 suppressed cells or is the increased F-actin polymerisation due to a de-stabilised WASH complex that has reduced inhibitory activity?

Response: We have performed these experiments and include them now in the revised manuscript. All subunits of the WASH complex bind to FAM21 to the same extent in Commd1-deficient cells (see Figure S4d). These are the same cells that have significant defects in CCC complex subunits and display exaggerated actin deposition, as other CCC deficiency states. Moreover, the mobility of the WASH complex under native electrophoresis conditions is not affected by COMMD3 deficiency (Figure 1d), which as shown elsewhere in the paper displays all the phenotypes of CCC deficiency.

14) Figure 6B: Given the suggestion that increased PI3P increases recruitment of WASH to endosomes, is it not surprising that inhibition of PI3P synthesis does not have a negative effect on WASH localisation in wild type cells?

Response: This observation probably reflects the fact that there are multiple mechanisms of recruitment, including for example retromer-dependent recruitment, which may not be affected by alterations in PI3P synthesis.

15) Figure 6D: Is the affect of the VPS34 siRNA in the COMMD3 KO cells statistically different to that in HeLa wild type cells? If so, is this a partial rescue of the phenotype? Does the collapsing of endosomes also rescue?

Response: In regard to Figure 6d, we believe that this is a partial rescue of the phenotype. This is likely due to the fact that we have no means to distinguish siVPS34 knockdown cells from those that did not receive siRNA. As far as the endosome collapsing phenotype, this was also quantified for this revision and we show that this part of the phenotype is also rescued (Figure S7d).

16) Figure 7A: Is this blot labelled correctly? Should the CCDC22 IP not be labelled as the FAM21 IP (and vice versa)?

Response: We appreciate the observation and apologize for the error. Indeed, this was mislabeled. In this revision we have removed any use of AcGFP-tagged MTMR2 (given concerns about its localization and behavior) and have replaced this panel with HA-tagged MTMR2. This also demonstrates CCDC22-MTMR2 binding and therefore the conclusion is unchanged.

17) In Supplementary Figure 7C are data derived from the same experiment?

Response: They are from the same experiment and were cropped to be able to provide different exposures in the same panel. These data have been removed from the revised manuscript.

18) Figure 7G: The imaging here does not look like the imaging in the paper the authors have referenced where the construct is either cytosolic or on Rab5 vesicles under serum starvation (no reticular phenotype is seen). Is the GFP tag on this construct affecting the localisation of the enzyme, and if so, can it be used in these specific experiments to draw any firm conclusion?

Response: We tried to stain for endogenous MTMR2 with a series of antibodies but found them not to be specific. As noted above, we have utilized other epitopes (HA and FLAG) for staining purposes. We find that short tags produce staining patterns that more closely resemble previous studies and have therefore eliminated the data with AcGFP due to concerns that the tag may be playing a role in the patterns observed. We are able to recapitulate the effects of MTMR2 phosphorylation on its localization that were previously reported. However, we have not seen that in our cell models, serum starvation produces a significant enough effect that could be useful for localization analysis. As noted elsewhere, we address this question by now showing that the CCC complex affects the phosphorylation status of MTMR2, which is key to its endosomal recruitment.

Reviewer #2 (Remarks to the Author):

This is a highly interesting manuscript that describes novel aspects of the interaction of the endosomal transmembrane protein cargo sorting complexes CCC, Retriever, Retromer and WASH and identifies the CC complex as a negative regulator of the WASH complex through regulation of MTMR2. Loss of CCC is shown to lead to an endosomal accumulation of actin polymerisation capable WASH complex. The MTMR2 mediated PI feedback loop is an important discovery, especially in the light of recent work that showed endosomal WASH retention is dependent on HRS, itself a PI3P binding protein (MacDonald et al. JCB2018).

It is interesting to see that an apparent overactivation of the WASH complex can stop recycling without significantly affecting retriever or retromer levels on the endosome. The authors need to point this out more clearly. While the possible conclusions are exciting, technical points need to be strengthened. In detail:

1) The CCC-retriever-retromer interactions have only recently been described and seem to be very complex, this is a very important aspect of this paper and the authors should quantify all their knock-out cell line complex protein blots in Fig.1 to add valuable information for the readership.

Response: We have provided this information as requested (see Figure 1e).

2) There are numerous experiments (for example Figs. 2c,d,e,f; 5C) where a factor whose endosomal association is changed in the knockout cells was used to quantify the association of other endosomal factors (for example WASH, EEA1). Can the authors please use endosomal protein factors that show unchanged association in their knockout cell lines or just use dextran loaded endosomes to quantify endosomal association of their proteins of interest.

Response: We tried to address this question using staining approaches first and found that dextran was not able to retain staining when cells would be fixed, permeabilized and treated for immunostaining. Additionally, we could not identify a marker that would fulfill the criteria set by the reviewer.

3) Line number: 103 Figure1a: This gel filtration experiments is very interesting. However, the results for the size of the WASH are somewhat at odds with previously published blue native (BN) gels results (Tyrrell et al. Pigment Cell Research) and do not match the molecular weight of a pentameric complex. It would be good if the authors could perform BN gel electrophoresis to reconcile their data with the published results. In addition, this would give a better idea about CCC and retromer complexes.

Response: We performed BN gels and western blots as suggested (see Figure 1b-d). These turned out to be very informative and we are grateful for the suggestion. First, we observed the same mobility pattern for WASH that was previously published. Second, this is indeed greater than the pentameric complex generated in vitro, although the explanation for this remains unclear. Please, keep in mind that characterizing the size or composition of the WASH complex is not our goal here; rather, our goal was to demonstrate that CCC and the WASH complexes are independent molecular assemblies. Third, this

approach allowed us to clarify a number of key questions about the CCC and retriever complexes (see above for additional comments). Finally, we decided to remove the gel filtration data since this approach has poor resolution between 2000 and 630 kDa which is the region occupied by most of these complexes.

4) Figure 1D can the authors explain why only VPS29 is one connected blot and the other proteins are not? Also, the VPS26C bands seem to be at an angle? Please include the original uncut blots as supplementary data to make it possible to interpret this experiment.

Response: All lanes were run together and the experiment was presented this way only to allow for different exposures to be presented in one consolidated panel, now shown in Figure S1d. To avoid any confusion, all inputs are shown as a separate panel, but figure S9 contains all the uncropped blots for further review.

5) In addition all blots in Fig.1 need MW markers.

Response: MW markers have been included throughout.

6) Line 165: ITGbeta1 recycling is also WASH dependent (Duleh&Welch Cytoskeleton 2012)
The assay in Fig. 2C does not necessarily show recycling of proteins. the intracellular accumulation could be caused by a.) increased endocytosis b.) defective degradation c.) decreased recycling. To make a statement about recycling all three possibilities need to be experimentally tested. Established biochemical assays should be used.

Response: All aspects suggested here, endocytosis, degradation and recycling, were investigated. These data are now included in Figure S2c-e. We found that endocytosis is unaffected upon loss of CCC components. We also found evidence of increased lysosomal degradation rather than reduced degradation, counter to the endosomal accumulation noted. Instead, we confirm through pulsed surface protein labeling experiments that the intracellular accumulation of integrin is indeed a recycling defect.

7) Line 300: if starvation results in increased MTMR2 activity it should result in reduced WASH and actin endosomes. The authors need to test this experimentally or remove the passage.

Response: We have amended this section extensively with the addition of new data. In particular, as noted above, we found that the CCC complex regulates MTMR2 phosphorylation status. In our experimental system using small epitope tags (instead of AcGFP), we did not observe an effect of serum starvation as previously reported. Please see our responses to reviewer 1 for additional details.

Reviewer #3 (Remarks to the Author):

Professors Billadeau and Burstein report CCC complex controls membrane protein recycling by

regulating PI(3)P levels within the endosomal compartment. The subject is interesting and timely, and the work is well done. This study is a well-presented paper with good assumptions and proof of it.

But it is doubtful whether it is appropriate to publish in a prestigious journal such as Nature Communications as a study of the mechanisms of already known phenomena, rather than a study of the discovery of new biological processes.

Billadeau and coworkers have earlier reported the discovery of an ancient and conserved multi-protein complex which orchestrates cargo retrieval and recycling (ref. Nat Cell Biol. 2017 Oct; 19(10): 1214–1225). They proposed the importance of SNX17 in retromer-independent cargo sorting pathway. In continuation of this work, in the present manuscript the authors uncover the critical role of CCC complex in endosomal trafficking of cargo proteins and report that endosomal PI(3)P regulation by the CCC complex controls membrane protein recycling.

Overall, it is an interesting and useful piece of work. The work is sound and has been carried out with great care taking care of all the possible questions. However, my major concern is, in the light of their earlier paper in Nat Cell Biol. 2017 whether the novelty level of this paper is good enough for Nature Communications. The following points should be addressed for publication.

Response: We respectfully disagree with the notion that the paper is incremental and not worthy of publication in Nature Communications. While the CCC complex has emerged as an important regulator of endosomal trafficking, its mechanism of action has remained unclear, and its activity above and beyond its interaction with retriever has not been previously elucidated. This study provides critical new insight into this conserved and essential regulator of endosomal biology and as such, we believe that the impact of this work in this field will be significant and is worthy of publication in this journal.

1. As in this study, studies that analyze protein interactions using LC-MS after IP often have problems associated with identification of false positive proteins. The authors should elaborate in detail how false positive proteins were removed from the list of identified proteins.

Response: The mass spectrometry data was analyzed using abundance based semi-quantitative methods, focusing on factors whose abundance was within 1 order of magnitude from the isolated baits. This was meant to identify components that would be likely to interact strongly with the utilized baits. The key interactions shown in the Venn diagram in Fig 1a were confirmed experimentally through co-IP experiments (shown in Suppl Figure 1a) and are in agreement with prior high throughput protein-protein interaction data (aggregate results presented in Table S2 and also in Suppl Fig 1c). Other putative interacting proteins provided in the supplementary tables (S1 and S2) have not been confirmed experimentally by us but are provided in this paper as they may spur additional investigation by other groups. We have clarified all these points in the revised manuscript.

2. Authors performed mass spectrometric identification of interacting proteins with CCC complex and retriever complex following HA-CDC93 or HA-VPS26C immunoprecipitation. In order to further clarify these results, it is recommended to perform and compare the same immunoprecipitation separation and mass spectrometry using another different CCC complex component and retriever complex component.

Response: As explained above, the data that we focus on for this paper was directly validated by independent co-IP experiments (shown in Figure S1a). In addition, we present as supplementary data the aggregate protein-protein interaction results of prior high throughput mass spec consortia, along with protein-protein interaction data from *Drosophila*, which are all in agreement with our mass spec results (Table S2).

3. The increase of the PI(3)P reporter signal (~ 2.5 fold) in endosomes from COMMD3 and CCDC93 KO cells, (Figure 5d, e) is the most important finding in this study as the authors highlighted it in the title. However, Figure 5e shows that there is a possibility that the PI(3)P concentration is interpreted as an overall increase due to the large deviation of the upward in COMMD3 KO. Statistically, a detailed description should be added to see if this deviation has been handled properly.

Response: That specific iteration appeared to have a non-Gaussian data distribution, but we believe it was simply the result of sample size and experimental variation. As an example, the data in 6a which extends the observations in 5e does not have non-normal distribution.

Reviewers' comments:

Reviewer #1 (Remarks to the Author):

A lot of work and effort has gone into this manuscript during the revision process, resulting in an improved paper. The emphasis of the manuscript has been changed to better reflect the story suggested and more evidence is given to the molecular mechanism of CCC and MTMR2 regulating endosomal PI3P levels.

Some points to address:

Figure 1e: How many independent experiments is the Western blot quantification from? Is it just the one blot shown or an average? If it is the latter a separate graph should be provided and perhaps with some statistical analysis. Quantification of WASH levels is important as see next point:

Line 239-240: 'Total WASH levels were increased'

It has not been shown that the total WASH levels have been increased and in fact figure 1e and line 147 may contradict this notion. Perhaps rephrase to indicate that endosomal associated WASH levels have increased to distinguish from total protein levels.

It would be nice to see the image channel splits for some of the imaging data, especially figures with phalloidin/WASH (e.g. 3a) because it is quite hard to see exactly what is going on with the merged colour images.

Reviewer #2 (Remarks to the Author):

The novel insights into CCC and retriever complex sizes and MTMR2 recruitment strengthen the manuscript considerably. The authors have answered most of my queries. I recommend to accept the manuscript if the following minor points are corrected/added:

-line 130-147. Parts of the blue native gels are labelled as Figure 2, whereas they are actually presented in Figure 1.

-FigS4D (and full blots in supplementary material), can the authors explain why they determined a 115kDa band to be Fam21 when they have previously estimated it to be around 230kDa (Gomez et al. *Dev. Cell* 17, 699–711, November 17, 2009; Gomez et al. *Mol Biol Cell*. 2012 Aug 15)? This is particularly interesting since the 230kDa band is absent from the shCOMMD1 immunoprecipitation.

- The authors have now added data to investigate receptor degradation and recycling. They show that COMMD3 KO results in loss of integrin expression, indicating an increased degradation rate. Increased sorting towards degradation and a block in recycling will both lead to endosomal retention of cargo in short pulse-chase experiments. Therefore, the pulse-chase experiment in FigS2e to investigate recycling is not convincing in its current form. Too really show a loss of recycling the endosomal integrin pool at 0min needs -at the very least- to be quantified and compared to the quantified 30min timepoint after the chase. A time-course of this experiment would make it even more convincing.

In addition, I do not understand how the intracellular staining in the HeLa WT cells can be brighter after the chase with unlabelled antibody. Are these images adjusted to same levels?

Alternatively, the authors could in my opinion remove any reference to a recycling defect of receptors in this manuscript and explore it in more detail in future studies.

Reviewer #3 (Remarks to the Author):

Authors responded successfully to the comments.

After minor revision, this manuscript can be published in the "Nature Communications".

1. It is recommended to change Fig 3F to a format suitable for showing individual data points and data distribution.

Point-by-point response to the second review

We are grateful for the positive review we received and also for the useful comments received. Below we address all remaining concerns, which we believe should make the paper acceptable for publication and we look forward to a positive outcome. We have highlighted in yellow any changes and addition to the text.

Reviewer #1 (Remarks to the Author):

A lot of work and effort has gone into this manuscript during the revision process, resulting in an improved paper. The emphasis of the manuscript has been changed to better reflect the story suggested and more evidence is given to the molecular mechanism of CCC and MTMR2 regulating endosomal PI3P levels.

Some points to address:

Figure 1e: How many independent experiments is the Western blot quantification from? Is it just the one blot shown or an average? If it is the latter a separate graph should be provided and perhaps with some statistical analysis. Quantification of WASH levels is important as see next point:

RESPONSE: The experiment was done 3 times, with some variations in lane loading and WB probing. The quantification shown in the figure represents the quantification in the blots shown and not the average of experiments because some of the blots were not as reliable in the other iterations.

Line 239-240: 'Total WASH levels were increased'

It has not been shown that the total WASH levels have been increased and in fact figure 1e and line 147 may contradict this notion. Perhaps rephrase to indicate that endosomal associated WASH levels have increased to distinguish from total protein levels.

RESPONSE: We have rephrased lines 147 and lines 239-240 to avoid any potential confusion about the points being made here. In this revision, we state that endosomal-associated WASH levels were increased (lines 239-240). The results of blue native gels are restated to highlight that WASH complex mobility is not affected in CCC deficient states, indicative of formation of the complex: we have removed any reference to abundance (line 147). We have avoided making any claims regarding total WASH protein quantification in the paper since the studies we performed were not intended to do so. However, we want to point out that quantification of WASH protein was performed by proteomic methods in primary hepatocytes (PMID: 29545368) showing a ~ 50% increase in abundance in COMMD1 deficiency, which is in agreement with the slight increase in WASH protein noted in Fig 1e for CCDC93 k/o (152%) and COMMD3 k/o (121%) cells.

It would be nice to see the image channel splits for some of the imaging data, especially figures with phalloidin/WASH (e.g. 3a) because it is quite hard to see exactly what is going on with the merged colour images.

RESPONSE: As requested, the images for Fig 3a have been split and are now provided in NEW Figure 3a.

Reviewer #2 (Remarks to the Author):

The novel insights into CCC and retriever complex sizes and MTMR2 recruitment strengthen the manuscript considerably. The authors have answered most of my queries. I recommend to accept the manuscript if the following minor points are corrected/added:

-line 130-147. Parts of the blue native gels are labelled as Figure 2, whereas they are actually presented in Figure 1.

RESPONSE: We apologize for this mistake, which we have now corrected.

-FigS4D (and full blots in supplementary material), can the authors explain why they determined a 115kDa band to be Fam21 when they have previously estimated it to be around 230kDa (Gomez et al. Dev. Cell 17, 699–711, November 17, 2009; Gomez et al. Mol Biol Cell. 2012 Aug 15)? This is particularly interesting since the 230kDa band is absent from the shCOMMD1 immunoprecipitation.

RESPONSE: FAM21 in mice is a 1334 amino acid long protein with a predicted molecular weight of 145 kDa. We have confirmed that this is indeed the dominant band by CRISPR, as shown here (not in the paper but confirmation of the mobility of the protein). In addition, we reviewed the western blot data shown in Fig S4D and noticed that the 115 kDa MW marker was slightly misplaced in the composite figure. We have corrected that in a new version of Fig S4D and also in Fig S9, and this shows more clearly that the mobility is indeed ~ 145 kDa.

- The authors have now added data to investigate receptor degradation and recycling. They show that COMMD3 KO results in loss of integrin expression, indicating an increased degradation rate. Increased sorting towards degradation and a block in recycling will both lead to endosomal retention of cargo in short pulse-chase experiments. Therefore, the pulse-chase experiment in FigS2e to investigate recycling is not convincing in its current form. To really show a loss of recycling the endosomal integrin pool at 0min needs -at the very least- to be quantified and compared to the quantified 30min timepoint after the chase. A time-course of this experiment would make it even more convincing.

RESPONSE: As suggested by the reviewer, we have now quantified the intensity of the integrin signal at 0 and 30 minutes for both the HeLa wild type and COMMD3-KO cells. The quantification is now shown in Figure S2e.

In addition, I do not understand how the intracellular staining in the HeLa WT cells can be brighter after the chase with unlabelled antibody. Are these images adjusted to same levels?

RESPONSE: We have gone back to the original images and realized the Time '0' image for HeLa wild type cells was not adjusted to the same brightness and contrast as the other three images. This has been corrected in this resubmission. We apologize for this oversight.

Alternatively, the authors could in my opinion remove any reference to a recycling defect of receptors in this manuscript and explore it in more detail in future studies.

Reviewer #3 (Remarks to the Author):

Authors responded successfully to the comments.

After minor revision, this manuscript can be published in the "Nature Communications".

1. It is recommended to change Fig 3F to a format suitable for showing individual data points and data distribution.

RESPONSE: This has been addressed in the current revision.